# Prickle isoform participation in distinct polarization events in the *Drosophila* eye

**Bomsoo Cho, Song Song, Joy Y. Wan, Jeffrey D. Axelrod**⊙*

Department of Pathology, Stanford University School of Medicine, Stanford, CA, United States of America

* jaxelrod@stanford.edu

**Data Availability Statement:** All relevant data are within the paper and its Supporting Information files.

**Funding:** This work was supported by NIH R01 GM097081 and NIH 5R35 GM13191402 to JDA

## Abstract

Planar cell polarity (PCP) signaling regulates several polarization events during development of ommatidia in the *Drosophila* eye, including directing chirality by polarizing a cell fate choice and determining the direction and extent of ommatidial rotation. The $pk^{sple}$ isoform of the PCP protein Prickle is known to participate in the R3/R4 cell fate decision, but the control of other polarization events and the potential contributions of the three Pk isoforms have not been clarified. Here, by characterizing expression and subcellular localization of individual isoforms together with re-analyzing isoform specific phenotypes, we show that the R3/R4 fate decision, its coordination with rotation direction, and completion of rotation to a final ±90° rotation angle are separable polarization decisions with distinct Pk isoform requirements and contributions. Both $pk^{sple}$ and $pk^{pk}$ can enforce robust R3/R4 fate decisions, but only $pk^{sple}$ can correctly orient them along the dorsal-ventral axis. In contrast, $pk^{sple}$ and $pk^{pk}$ can fully and interchangeably sustain coordination of rotation direction and rotation to completion. We propose that expression dynamics and competitive interactions determine isoform participation in these processes. We propose that the selective requirement for $pk^{sple}$ to orient the R3/R4 decision and their interchangeability for coordination and completion of rotation reflects their previously described differential interaction with the Fat/Dachsous system which is known to be required for orientation of R3/R4 decisions but not for coordination or completion of rotation.

## Introduction

PCP signaling controls the polarization of cells within the plane of an epithelium, orienting asymmetric cellular structures, cell divisions and cell migration. In flies, PCP signaling controls the orientation of hairs on the adult cuticle, chirality and orientation of ommatidia in the eye, orientation of cell divisions, and related processes in other tissues. While much mechanistic understanding of PCP signaling derives from work using *Drosophila* as a model system, medically important developmental defects and physiological processes in vertebrates are also under control of PCP signaling, motivating considerable interest in studying PCP both in *Drosophila* and in vertebrate model systems. Defects in the core PCP mechanism result in a range of developmental anomalies and diseases including open neural tube defects (reviewed in [1, 2]), conotruncal heart defects (reviewed in [3]), deafness (reviewed in [4]), situs inversus and

(https://www.nigms.nih.gov). The funders had no role in study design, data collection and analysis, decision to publish, or preparation of the manuscript.

**Competing interests:** The authors have declared that no competing interests exist.

heterotaxy (reviewed in [5]). PCP is also believed to participate in both early and late stages of cancer progression (reviewed in [6, 7]) and during wound healing [8, 9]. PCP polarizes skin and hair (e.g. [10, 11]) and the ependyma (e.g. [12]), and participates in renal tubule development [13]. The PCP component Pk, though likely not the PCP pathway, is mutated in an epilepsy-ataxia syndrome [14, 15]. Mutations in 'global' PCP components have been associated with a human disorder of neuronal migration and proliferation [16].

Work in *Drosophila* indicates that at least two molecular modules contribute to PCP. The core module acts both to amplify molecular asymmetry, and to coordinate polarization between neighboring cells, producing a local alignment of polarity. Proteins in the core signaling module, including the serpentine protein Frizzled (Fz), the seven-pass atypical cadherin Flamingo (Fmi; a.k.a. Starry night), the 4-pass protein Van Gogh (Vang; a.k.a. Strabismus), and the cytosolic/peripheral membrane proteins Dishevelled (Dsh), Diego (Dgo), and the PET/Lim domain protein Prickle (Pk) adopt asymmetric subcellular localizations that predict the morphological polarity pattern such as hairs in the fly wing (reviewed in [17]). These proteins communicate at cell boundaries, recruiting one group to the distal side of cells, and the other to the proximal side, through the function of an incompletely understood feedback mechanism, thereby aligning the polarity of adjacent cells. A global module serves to convert tissue level expression gradients to asymmetric subcellular Fat (Ft)—Dachsous (Ds) heterodimer localization, and provides directional information to the core module. It consists of the atypical cadherins Ft and Ds that form heterodimers which may orient in either of two directions at cell-cell junctions, and the Golgi resident protein Four-jointed (Fj). Fj phosphorylates the ectodomains of both Ft and Ds to make Ft a stronger ligand, and Ds a weaker ligand, for the other. As Fj and Ds are expressed in gradients across tissues, the result is the conversion of transcriptional gradients to subcellular gradients, producing a larger fraction of Ft-Ds heterodimers in one orientation relative to the other (reviewed in [18, 19]). Other less well defined sources of global directional information appear to act in partially overlapping, tissue dependent ways [20].

In the *Drosophila* eye, each ommatidium contains 8 unique photoreceptor cells. During eye morphogenesis in the third instar larval stage, photoreceptor cell fates are sequentially specified as cells become part of ommatidial clusters in a stereotypical fashion behind the morphogenetic furrow as it sweeps toward the anterior of the eye disc [21]. Maturing ommatidia in the dorsal and ventral sides of the eye field rotate 90 degrees in opposite clockwise or counter-clockwise directions to produce the adult dorso-ventral mirror image ommatidial arrays (Fig 1A and 1B). Two R3/4 precursor cells each have the potential to become either R3 or R4, and this cell fate decision, which occurs during the five cell (R8, 2, 5, 3, and 4) stage, establishes the more polar of the two precursors as R4 and the more equatorial of the two as R3. The decision to rotate clockwise or counter-clockwise is coordinated with the cell fate decision, such that rotation initiates toward the polar R4 side, positioning the R3 to the anterior and the R4 to the posterior side of the adult eye (Fig 1A). The rate and extent of cluster rotation is also tightly regulated so that initial fast (row 5 to row 9) and then slower (row 9 to row 15) rotation is synchronized as it proceeds to a final ±90˚ rotation angle. A consequence of sequential initiation and differentiation behind the furrow is that ommatidia in an individual disc effectively form a time series, where rows closest to the furrow are youngest, and those in rows of increasing distance from the furrow are progressively older [21].

During ommatidial morphogenesis, PCP signaling is implicated in control of each of these events [22–27]. Mutation of most core PCP components disrupts the R3/R4 cell fate decision, giving rise to a portion of ommatidia with reversed R3 and R4 specification or with symmetric or unspecified R3/4 fates (Fig 1D and description). In addition to the defects related to the cell fate decision, ommatidia in core PCP mutants fail to rotate properly, displaying various extents

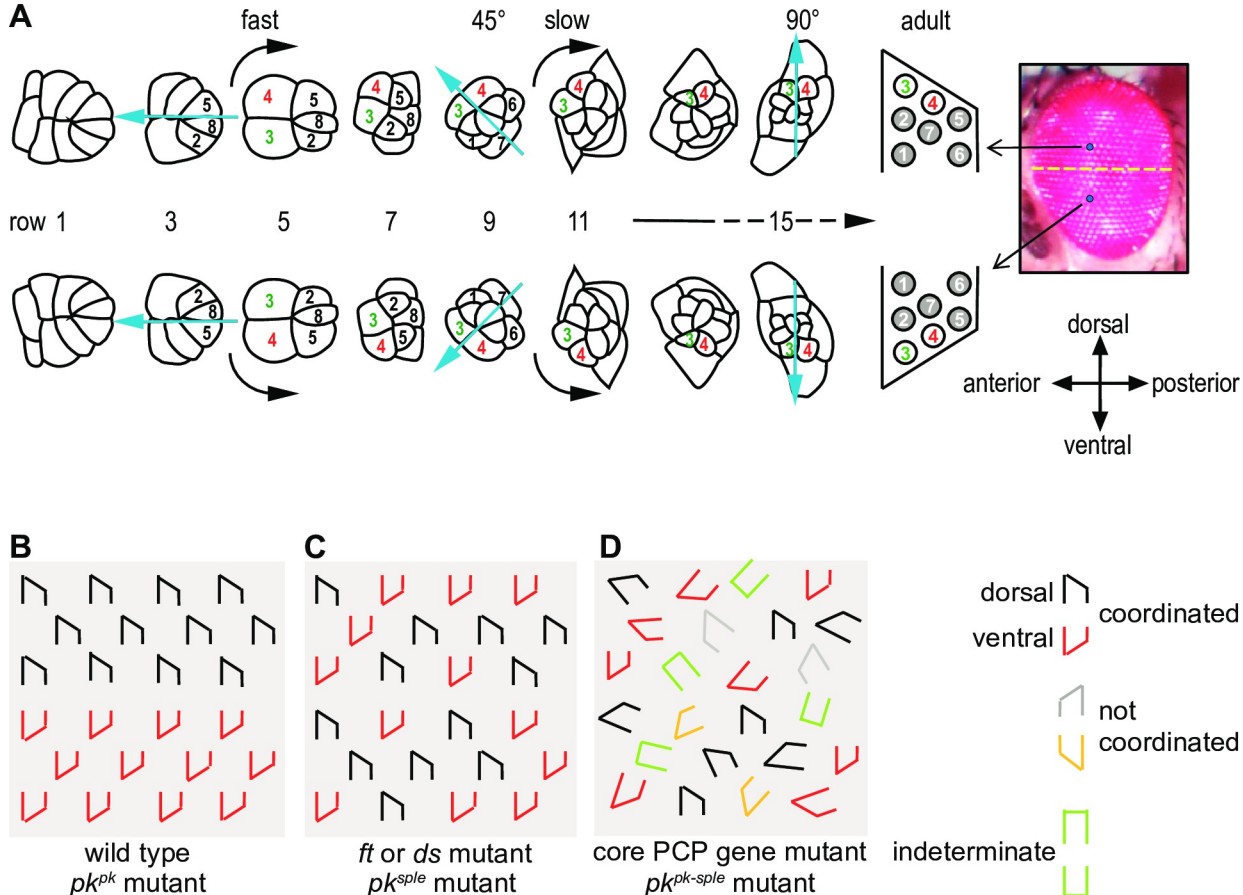

**Fig 1. Ommatidial maturation and PCP mutant phenotypes.** A. Cells are recruited into photoreceptor and other cell fates stereotypically as ommatidia develop. Photoreceptor cell numbers are shown. Simultaneously, the clusters rotate in opposite directions on either side of the equator in an initial fast and then a slow phase. Each row is approximately two hours older than the subsequent row. B. Schematic of ommatidia in a wildtype and a $pk^{pk}$ mutant eye. Ommatidia have opposite chirality in the dorsal and ventral hemispheres. C. In *ft*, *ds* or $pk^{sple}$ mutant eyes, dorsal and ventral type ommatidia are intermixed and no equator is apparent. D. In core PCP mutant eyes, including $pk^{pk-sple}$ eyes, some ommatidia fail to distinguish R3 and R4 (indeterminate), dorsal and ventral type ommatidia are intermixed, and ommatidia rotate in the correct or incorrect direction and to varying extent, or not at all.

of rotation from 0 to ±90° degrees (Fig 1D and description). Furthermore, core PCP mutations cause failure to coordinate rotation direction with the R3:R4 orientation, resulting in R4 being positioned at the anterior rather than the posterior side of the eye field (Fig 1D and description).

The *pk* locus encodes three isoforms that result from differential splicing of alternative N-terminal sequences. Two differentially expressed isoforms of Pk, Pk$^{prickle}$ (Pk$^{pk}$) and Pk$^{spiny-legs}$ (Pk$^{sple}$), are known to control PCP in various *Drosophila* tissues, while a third isoform, Pk$^{m}$, was until now thought to be only expressed in embryos and has not been studied as no selective mutant was available [28, 29]. In addition to the requirement for either isoform to generate asymmetric localization of the core PCP components, these isoforms have been proposed to differentially determine the direction in which core PCP signaling responds to information provided by the Ft/Ds/Fj system. Pk$^{sple}$ is thought to directly interact with components of the Ft/Ds/Fj system to orient core PCP signaling [30, 31], whereas in Pk$^{pk}$-predominant tissues, interpretation of the Ft/Ds/Fj directional signal occurs through indirect means [32–35]. While it was originally believed that one or the other of these isoforms is predominant in each tissue,

more nuanced analyses now suggest that dynamic spatiotemporal expression determines their contributions, at least in the wing blade and anterior wing margin [36].

In the eye, *pk^pk* mutant ommatidia are patterned normally (Fig 1B), whereas *pk^sple* mutation causes frequent reversals of R3/4 polarity such that dorso-ventral mirror symmetry is disrupted [22, 25] a phenotype resembling that seen in *ft* or *ds* mutants (Fig 1C) [37, 38]. The simple interpretation is that polarity in the eye depends on Pk^sple and not Pk^pk. However, unlike in other core PCP mutants or *pk^pk-sple* mutants [22, 39], in *pk^sple* mutants the direction of ommatidial rotation is properly coordinated with R3/4 cell fate, whether correctly or incorrectly specified (R4 uniformly positioned to the posterior), and rotation proceeds normally through 90° [22, 25, 40]. Thus, while *pk^pk* is not needed, in the absence of functional *pk^sple*, *pk^pk* rescues the rotation and coordination defects observed in *pk^pk-sple* mutant eyes. These observations suggest that the presence of either Pk^pk or Pk^sple in the absence of the other might be sufficient to both coordinate the direction of rotation to the R3/R4 fate decision and to control the extent of ommatidial rotation (compare Fig 1C to 1D). The possible contributions of Pk^pk and Pk^sple to coordinating rotation direction with the fate decision and to controlling extent of rotation have not been described. Furthermore, how Pk^pk and Pk^sple might interact with the Ft/Ds/Fj system in these processes is not known.

Two cells selected at the anterior of each ommatidium become bipotent prospective R3/R4 cells. They then adopt their R3 and R4 cell fates through a Notch (N) mediated competition [23, 24]. In wildtype, invariably the more equatorial of the two becomes R3 and the more polar becomes R4. The N competition is biased by PCP signaling between the two cells, with higher Fz and Dsh levels accumulating on the R3 side and higher Vang levels accumulating on the R4 side of the prospective R3:R4 junction [27]. This oriented PCP signaling promotes greater Delta activity in the equatorial R3 and N activity in the polar R4. The R3/R4 fate decision is made no later than row 5, as a N-specific reporter is detected in R4 beginning at this time [23]. Consistent with the role of PCP signaling in biasing the N competition, asymmetric distribution of Vang [26], Fz [27], Fmi and Dsh [38] is evident by row 5.

While competition and competitive interactions between Pk^pk and Pk^sple in controlling core PCP signaling have been posited for hair and bristle cells in some tissues, how PCP signaling and the potential interplay of Pk isoforms control the various steps in ommatidial patterning has not been elucidated. Here, an analysis of Pk isoform participation in ommatidial development allows us to define the R3/R4 cell fate decision, ommatidial rotation, and coordination of rotation direction to the R3/R4 decision as separable steps with distinct contributions of Pk isoforms in core signaling. We propose that a requirement for Ft/Ds/Fj signaling in orienting the R3/R4 fate decisions demands Pk^sple participation, whereas the absence of this requirement in coordination and completion of rotation allows the interchangeable participation of either Pk^pk or Pk^sple.

## Results

### The R3/R4 cell fate decision

We examined how isoform expression relates to prior observations suggesting that the Pk^sple isoform participates in the R3/R4 fate decision. *pk^pk* mutant adult eyes show normal chirality. In contrast, *pk^sple* mutant eyes display a mixture of dorsal and ventral ommatidia throughout the eye with no clear equator [22, 25], indicating that rather than selectively specifying the polar R3/R4 precursor as R4, the choice is made randomly. An antibody recognizing a common domain in all Pk isoforms (Pk[C]) detects Pk at apico-lateral junctional membranes in most cells immediately behind the furrow and enriched signal in maturing ommatidia from row 3 to at least row 17 (S2 Fig). Examination of endogenously tagged isoforms of Pk^sple and

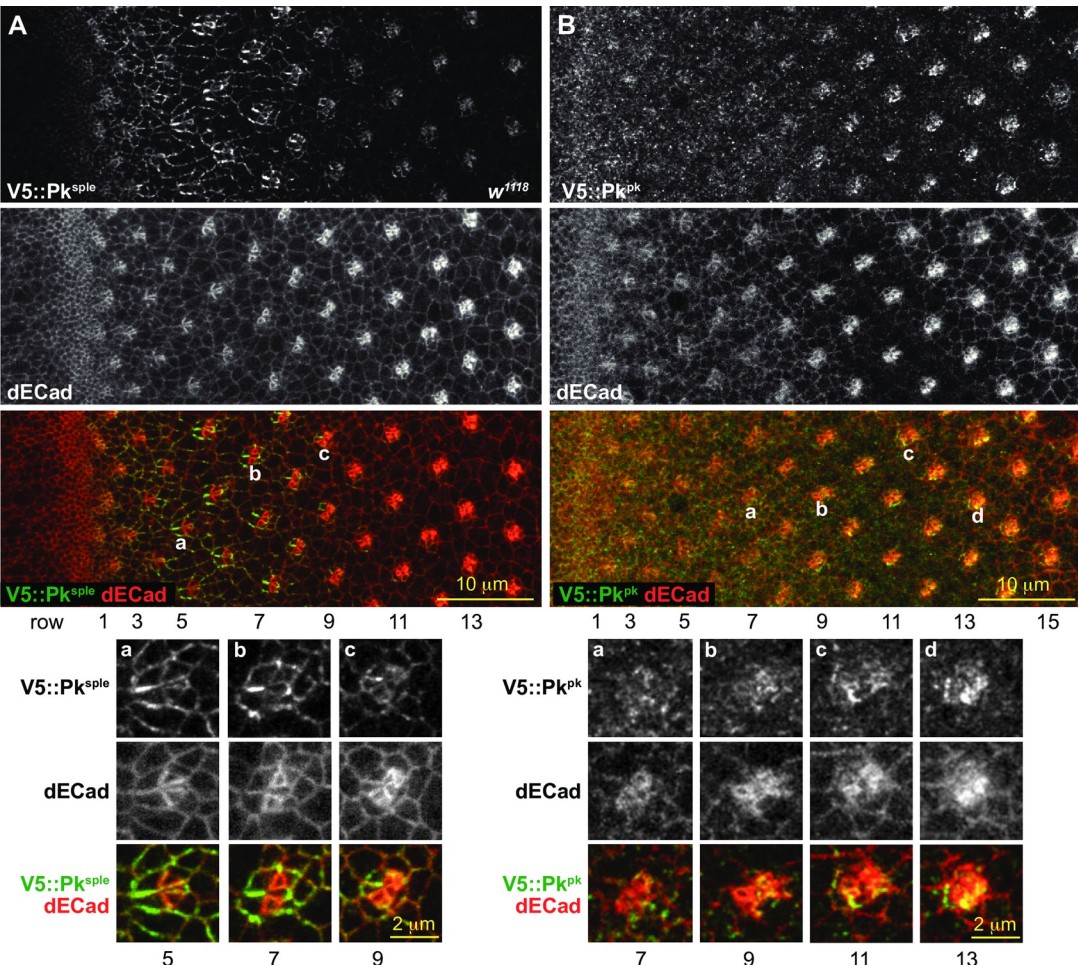

**Fig 2. Pk^pk and Pk^sple apico-lateral junctional expression in wildtype eyes.** A. Endogenously tagged V5::Pk^sple stained for V5 (green in merge) and dECad (red in merge), showing V5::Pk^sple enriched at apico-lateral junctions of early ommatidia. Aa-c. Enlarged images of ommatidia from rows 5, 7 and 9 as marked. B. Endogenously tagged V5::Pk^pk stained for V5 (green in merge) and dECad (red in merge), showing V5::Pk^pk enriched at apico-lateral junctions of late ommatidia. Ba-d. Enlarged images of the ommatidia from rows 7, 9, 11 and 13 as marked.

Pk^pk reveals that Pk^sple is selectively expressed and enriched at apico-lateral junctions of early ommatidia, when R3 and R4 are specified, and then starts to decline between rows 7 and 9 (Figs 2A and S2A). In contrast, very little Pk^pk is expressed during the R3/R4 fate decision, but Pk^pk becomes enriched at apico-lateral junctions from row 9 and later (Figs 2B and S2B). In rows 4–9, Pk^sple shows asymmetric distribution in R3/R4 pairs in a pattern closely resembling that of Vang (Fig 3B). Indeed, the Vang apico-lateral junctional localization depends on the presence of Pk^sple: Vang apico-lateral junctional localization is lost in a *pk^pk-sple* mutant (Fig 3A), and in a *pk^sple* mutant it only appears later with expression of Pk^pk (see below; Figs 3, 4, S3 and S4). These observations are consistent with the R3/R4 fate decision depending on participation of Pk^sple for the PCP-dependent biasing of the N signaling event.

Though the wildtype appearance of *pk^pk* mutant eyes suggests no role for Pk^pk in the R3/R4 fate decision, the difference between the *pk^sple* phenotype and that of *pk^pk-sple*

(and other core PCP mutants) suggests otherwise. In addition to both dorsal and ventral type ommatidia, the phenotype of *pk^pk-sple* and other core PCP mutants displays numerous indeterminate ommatidia resulting from a failure to distinguish R3 and R4 (either R3:R3, R4:

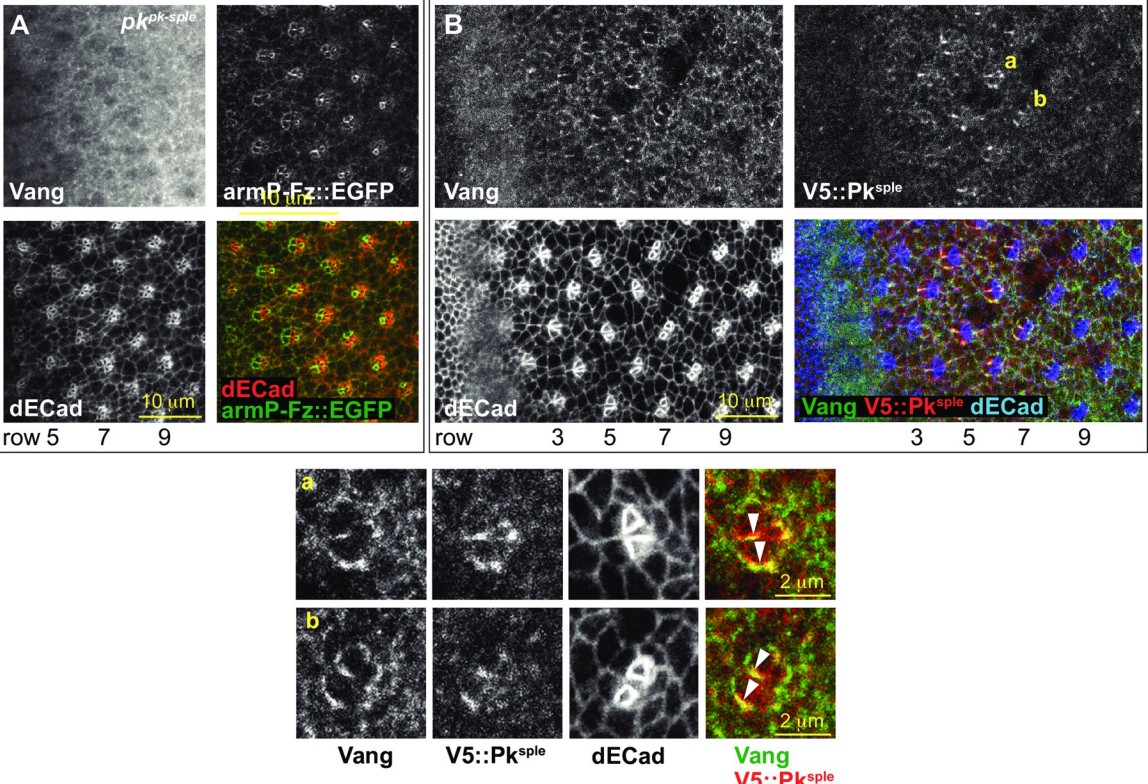

**Fig 3. The relationship between Vang apico-lateral localization and Pk.** A. In a $pk^{pk\text{-}sple}$ mutant eye, Vang fails to localize at apico-lateral junctions. B. Vang co-localizes with V5::Pk$^{sple}$ at apico-lateral junctions of early ommatidia. Vang (green in merge), V5::Pk$^{sple}$ (stained for V5; red in merge) and dECad (blue in merge). Ba-b. Enlarged images of ommatidia from panel B rows 6 and 7 as marked, showing co-localization and asymmetry of Vang and V5::Pk$^{sple}$ at the equatorial sides of R3 and R4 (arrowheads).

R4, or undifferentiated) that are only very rarely present in $pk^{sple}$ mutant eyes (Fig 1) [25]. In $pk^{sple}$ eyes, failed R3/R4 decisions therefore appear to be prevented by the presence of an intact $pk^{pk}$ allele. However, in wildtype, expression of endogenously tagged Pk$^{pk}$ apico-lateral junctional expression is not apparent until after the R3/R4 fate decision is made (Fig 2). We therefore examined expression of Pk$^{pk}$ in the $pk^{sple}$ mutant. We were unable to create viable endogenously tagged $pk^{pk}$ on a $pk^{sple}$ chromosome for unknown reasons, so we instead analyzed expression detected by the common antibody that is predicted to detect Pk$^{pk}$ in a $pk^{sple}$ mutant fly. Whereas apico-lateral junctional expression of endogenously tagged Pk$^{pk}$ in wildtype ommatidia becomes prominent at around row 9 (Fig 2), in $pk^{sple}$, the Pk common antibody detects localization at apico-lateral junctions in ommatidia beginning at row 5 (Fig 4). Because endogenous Pk$^{pk}$ apico-lateral expression is observed slightly later in wildtype eyes, and because a wildtype $pk^{pk}$ allele can rescue the indeterminate decisions in $pk^{sple}$ mutants [27], we presume that the Pk[C] antibody is detecting Pk$^{pk}$, though we cannot strictly rule out that it is detecting Pk$^{m}$. Furthermore, it is possible that the earlier detection of the signal by the common antibody in the $pk^{sple}$ mutant reflects a level of expression below the detection threshold of the endogenously tagged V5::Pk; however, we believe this is unlikely because our experience suggests high sensitivity with the V5 antibody. Therefore, regardless of whether absence of Pk$^{sple}$ allows earlier access of Pk$^{pk}$ to participate in PCP signaling, a sufficient amount of Pk$^{pk}$ (or less likely Pk$^{m}$) is present in apico-lateral junctional complexes in time to largely

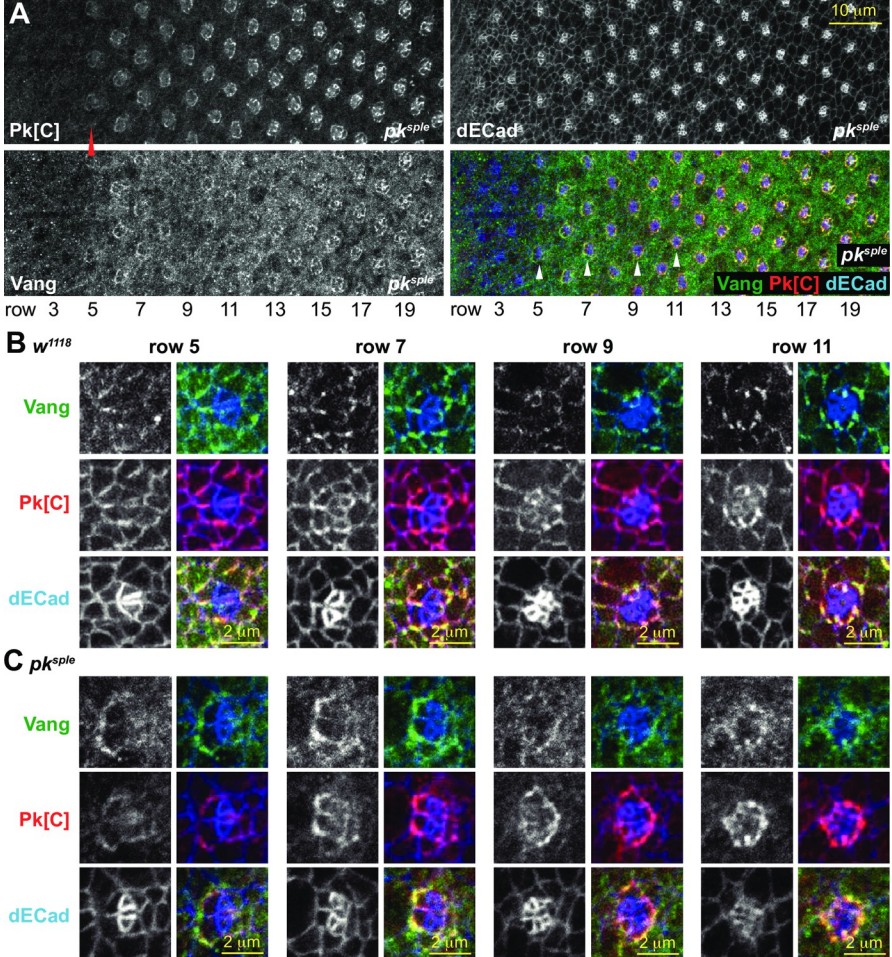

**Fig 4. Vang and Pk in a *pk^sple^* mutant eye.** A. A *pk^sple^* mutant eye stained with the Pk[C] common antibody (detects mostly Pk^Pk^) and a Vang antibody. Note that Pk^Pk^ is weakly detected at apico-lateral junctions beginning at row 5, somewhat earlier than in control (*w^1118^*), and Vang is detected in a similar pattern. Enlarged images of ommatidia from a control (*w^1118^*; B) and a *pk^sple^* (C) mutant eye from rows 5, 7, 9 and 11 as marked in A and S3 Fig (for B). Vang appears at apico-lateral junctions somewhat later, concomitant with expression of Pk^pk^, in the *pk^sple^* mutant eye. Vang (green in merge), Pk[C] (red in merge) and dECad (blue in merge). Asymmetry of Pk^pk^ and Vang localization is often delayed, as in this row 7 ommatidium.

prevent the failed R3/R4 fate decisions seen in other core PCP mutants. Though apico-lateral junctional Pk in the *pk^sple^* mutant is first detected in row 5, its expression is modest until several rows later, strengthening only after the time that the R3/R4 decision is made in wildtype (row 5). Consistently, we observe that ommatidial maturation is somewhat asynchronous in *pk^sple^* mutants. While some ommatidia show asymmetry of Vang localization by row 5, others remain symmetric as late as row 7 (S5 Fig). Though early asymmetry is often not apparent, R4 cells are nearly always distinguishable in older ommatidia (Fig 4). Accumulating apico-lateral junctional Pk^pk^ expression might therefore facilitate PCP signaling, albeit sometimes delayed, to enable R3/R4 decisions in ommatidia that have not yet been determined by the N competition alone.

In prior work, the Ft/Ds/Fj system has been shown to direct the mirror image polarity of the eye: equatorial-polar gradients of Ds and Fj are proposed to establish asymmetric

orientation of Ft-Ds heterodimers that in turn orient the core PCP signal in mirror image equatorial-polar directions [37, 38]. Similarly, the Ft/Ds/Fj system provides directional information in polarization of hairs on the wing and abdomen and bristles on the anterior wing margin [36, 41, 42]. In wing and abdomen, directional information is differentially interpreted by the core PCP system depending on the Pk isoform that is predominant, such that substituting isoforms reverses the normal polarity of hairs and bristles [30–33]. It is thought that binding of Pk$^{sple}$ to asymmetrically oriented Ft-Ds heterodimers couples Pk$^{sple}$-dependent core polarization to the Ft/Ds/Fj signal, while Ft-Ds directed polarized microtubule trafficking of vesicles carrying Fmi, Fz and Dsh loosely couples Pk$^{pk}$-dependent PCP signaling to the Ft/Ds/Fj system in the opposite orientation [17, 19].

By analogy to the wing, in $pk^{sple}$ mutant eyes, if Pk$^{pk}$ substitutes for Pk$^{sple}$ in the R3/R4 decision, one might expect that the population of ommatidia deciding R3/R4 fate based solely on N would be random, but those whose decision is rescued by Pk$^{pk}$ would be reversed, leading to an excess of reversed ommatidia. However, R3/R4 decisions are random in $pk^{sple}$ mutant eyes [25]. We hypothesize two non-exclusive possible explanations: (i) unlike in other tissues, there may be no coupling between the Ft/Ds/Fj system and Pk$^{pk}$-dependent core PCP signaling, or (ii) delayed R3/R4 decisions may occur too late to be subject to an earlier acting biasing signal. Evidence for hypothesis (i) comes from the prior observations that in $pk^{pk-sple}$ mutant eyes, timely induced expression of Pk$^{pk}$ yields approximately randomized rather than reversed polarity [43], and that ectopically expressed Pk$^{pk}$ co-localizes with the endogenous Pk$^{sple}$ on the equatorial side of R4 rather than competing with an opposite orientation to reverse the polarity of R4 [31].

We also tested whether the R3/R4 decision is time limited while under the normal Pk$^{sple}$ control (hypothesis ii) by delaying activation of the PCP signal. We delayed expression of Vang in a *vang* mutant eye by expressing it under GMR-GAL4 control and assayed rescue of the *vang* mutant phenotype (Fig 5). GMR-GAL4 driven Vang restores decisions to most ommatidia whose R3/R4 decisions would otherwise be indeterminate (the intermediate ommatidia were reduced to 10.97% in *GMR>vang* flies compared to 29.42% in *GMR>w$^{RNAi}$* flies; 8 adult eyes for each analyzed), but these decisions are random like those in $pk^{sple}$ mutants (numbers of dorsal and ventral form of ommatidia were roughly equal (48% vs. 52% respectively), 8 adult eyes). While the timing is not precisely defined by this experiment, if, as expected, the GMR-GAL4 driven Vang is restored while Pk$^{sple}$ is still expressed, we would conclude that a late R3/R4 decision is not responsive to the Ft/Ds/Fj system. Thus, though not definitive, these observations suggest that Pk$^{sple}$ but not Pk$^{pk}$ is coupled to Ft/Ds/Fj in the eye and that there is an early, time limited window for responsiveness of the Pk$^{sple}$-dependent R3/R4 fate decision to Ft/Ds/Fj.

## Ommatidial rotation

We next examined how PCP signaling regulates ommatidial rotation. At approximately the time the R3/R4 fate decision is made, the developing ommatidial cluster begins to rotate in the direction of R4, undergoing a roughly 90˚ rotation so that R4 ends up posterior to R3 [21]. Rotation depends on PCP, as in core PCP mutants, ommatidia with or without specified R3: R4 pairs often undergo partial or no rotation and may progress in either direction [21]. In contrast, in *ft*, *ds* and *fj* mutants, ommatidia rotate correctly through 90˚ with respect to the R3:R4 decision, whether equatorial:polar or polar:equatorial [37, 38] (Fig 1).

Core PCP signaling is required from the beginning of the rotation process, as rotation is delayed in *vang* [27, 44] and $pk^{sple}$ mutants, often initiates in the wrong direction, and in some ommatidia does not occur at all. It is also likely required throughout rotation, as rotation does

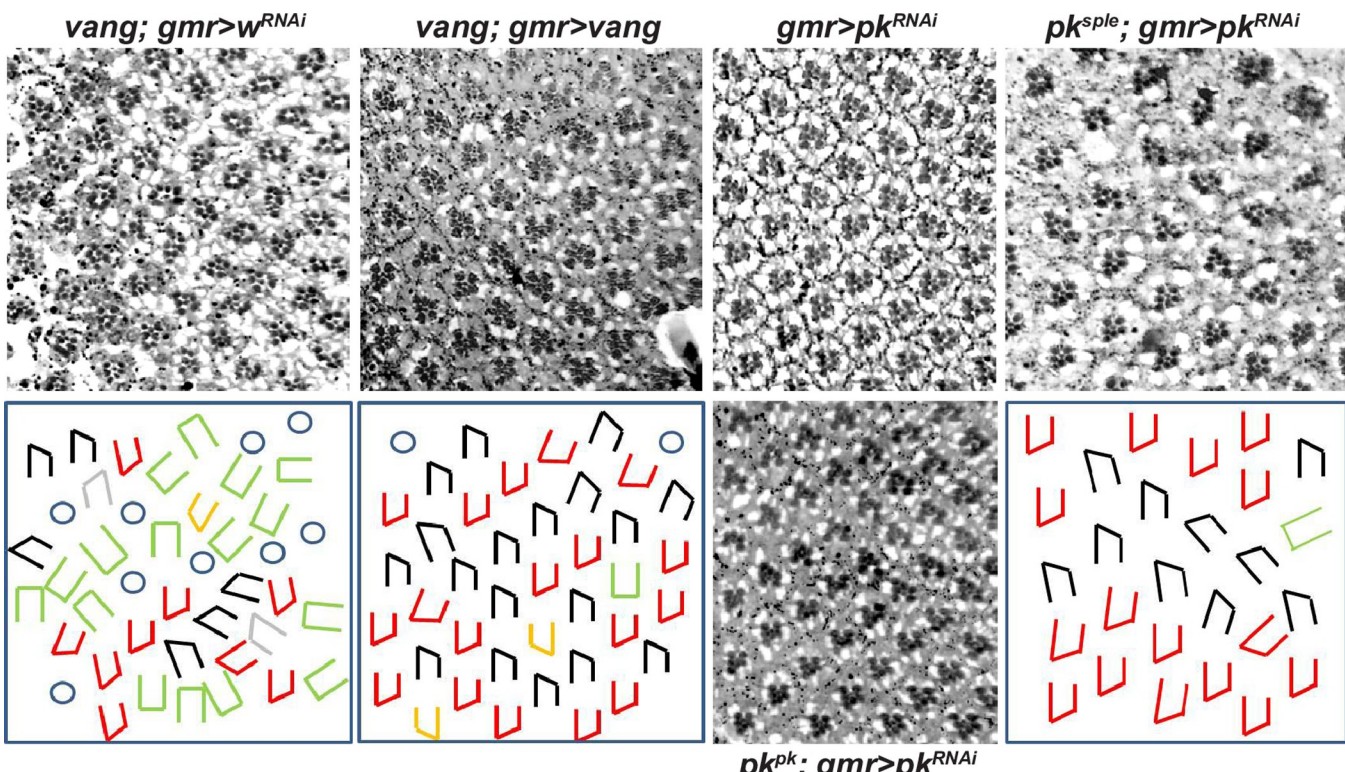

**Fig 5. Delayed restoration of *vang* or knockdown of *pk^pk^*.** Expression of control *w^RNAi^*, *vang*, or *pk^RNAi^* was driven by GMR-GAL4 in mutant or wild type backgrounds as indicated. GMR-GAL4 expression becomes significant by row 7 as judged by loss of Pk[C] signal upon GMR-GAL4-driven knockdown by *pk^RNAi^* (S6 Fig). Restoration of vang expression corrects most indeterminate R3/R4 decisions, and allows most rotations to complete to 90˚, but rotation direction is uncoordinated with the R3/R4 decision. In a *pk^sple^* mutant eye, intermixed dorsal and ventral ommatidia rotate to 90˚, but delayed knockdown of remaining *pk* isoforms (mostly *pk^pk^*) causes under-rotation of some ommatidia.

not proceed to completion in core mutants including *vang*, *pk^pk-sple^*, *fz* and *fmi* [22, 44–46] (but note the important exceptions of *pk^sple^* and *pk^pk^* discussed below). An ongoing requirement for PCP signaling during rotation is also supported by the finding that later partial knockdown of all Pk isoforms by GMR-GAL4 driven *pk* RNAi in a *pk^sple^* background prevents rotation of some ommatidia from proceeding to completion (10.21% ommatidia failed to complete rotation (8 adult eyes); compared to less than 2% failure of complete rotation in *pk^sple^* mutant itself (8 adult eyes)) (Figs 5 and S6), and by the related observation that delayed restoration of Vang in a *vang^A3^* mutant background (driven by GMR-GAL4) corrects what would otherwise be partial rotations to complete ones. In contrast to core signaling, the Ft/Ds/Fj system is not required for rotation, as ommatidia in mutants affecting this system rotate normally to completion [37, 38].

As described earlier, Pk^sple^ is the predominant isoform in ommatidia when rotation begins, but its expression soon declines and Pk^pk^ becomes the predominant isoform in the apico-lateral junctions of older ommatidia (Fig 2). Early core PCP signaling required for rotation apparently therefore incorporates Pk^sple^, while later signaling apparently incorporates Pk^pk^. One inference from this observation is that both isoforms can fulfill the same role in core PCP signaling during rotation. Confirmatory evidence for this idea arises from analyses of *pk^sple^* and *pk^pk^* mutants. If Pk^pk^ is selectively apico-laterally enriched in older ommatidia, what sustains later rotation in *pk^pk^* mutants? We found that in *pk^pk^* mutant eye discs, apico-lateral Pk^sple^

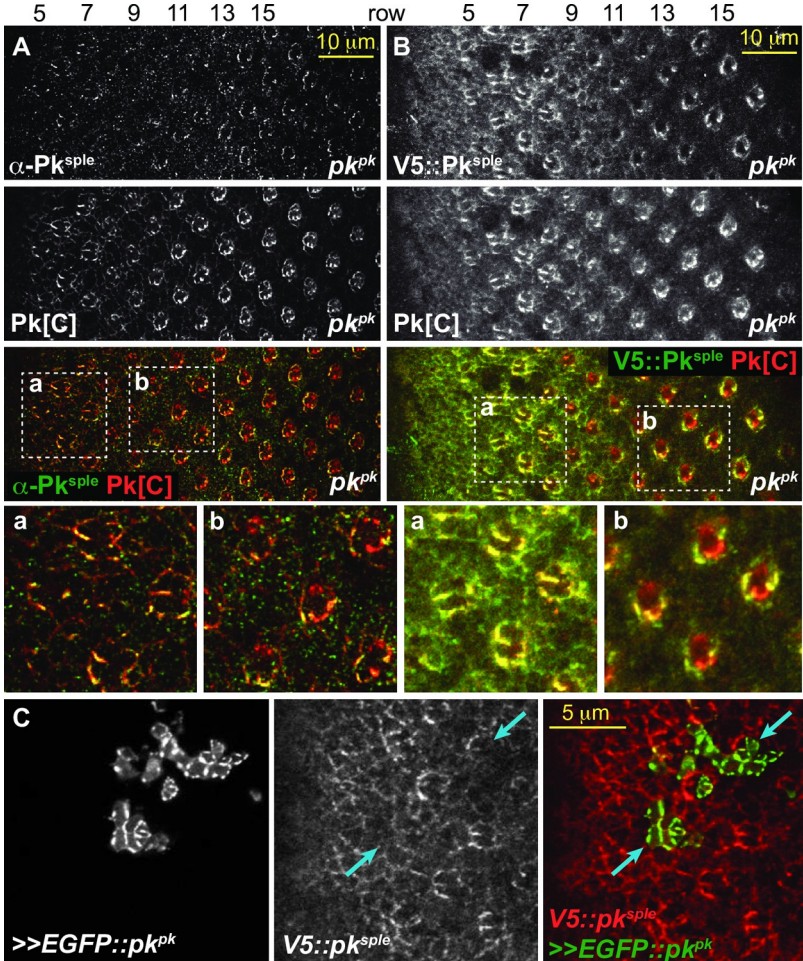

**Fig 6. Pk^sple replaces Pk^pk in apico-lateral domains of older pk^pk mutant ommatidia.** A. A $pk^{pk}$ mutant eye stained with anti-Pk^sple and the common Pk[C] antibody. The common antibody recognizes Pk^sple and Pk^m in this condition. Pk^sple is detected in apico-lateral junctions of early ommatidia as in wild type (rows 5–7), then declines briefly and is again enriched in apico-lateral junctions of older ommatidia (row 9 and later). B. A $pk^{pk}$ mutant eye expressing endogenously tagged V5::Pk^sple and stained with anti-HA and the common Pk[C] antibody. As in A, endogenously tagged V5::Pk^sple is enriched in apico-lateral junctions of older ommatidia where Pk^pk is predominant in control eyes. Aa-b and Ba-b show magnified regions marked in A and B. C. High level clonal overexpression of EGFP-tagged Pk^pk competes and displaces V5::Pk^sple from ommatidia in rows 3–5. Blue arrows indicate ommatidial clusters where ectopic EGFP::Pk^pk reduces apico-lateral V5::Pk^sple signal.

expression begins to decline in rows 7–9, but rather than continuing to decline to extinction, rises again, replacing the missing apico-lateral Pk^pk expression in older rows with Pk^sple (Fig 6). Conversely, in $pk^{sple}$ mutants, apico-lateral Pk^pk expression starts somewhat earlier than in wildtype and remains present, supporting complete, though slightly delayed, rotation (S5 Fig). Thus, while Pk^pk normally replaces Pk^sple to support rotation, Pk^pk and Pk^sple can each localize apico-laterally and support rotation to completion in the absence of the other. We point out that the absence of input from Ft/Ds/Fj to rotation appears to facilitate this interchangeability between Pk^pk and Pk^sple by eliminating what might otherwise be differential interactions that would cause the two isoforms to behave differently.

Curiously, we observed that while Pk^pk is the predominant isoform in older ommatidia, the Pk[C] common antibody detected signal in R4 that could not be accounted for by

endogenously tagged Pk$^{pk}$ or Pk$^{sple}$ (S7 and S8 Figs). Instead, we found that Pk$^m$, the isoform previously reported to be restricted to the embryo, is highly expressed in older R4 cells. Eyes in CRISPR induced $pk^m$ mutant flies are morphologically normal (S8D Fig), but we do not rule out potential phenotypes if combined with mutations of other isoforms.

Our results show that ommatidial rotation can occur in the absence of intact PCP signaling, but that timely initiation, rotation in the correct direction, and conclusion at 90˚ all depend on PCP signaling. Apico-lateral junctional localization of either Pk$^{pk}$ or Pk$^{sple}$ alone can support the PCP rotation function, and no interpretation of Ft/Ds/Fj signaling is involved in the rotation event.

## Interaction between isoforms

In wing development, polarization of hairs depends on Pk$^{pk}$ participation in PCP signaling. Subsequently, Pk$^{sple}$ becomes the predominant isoform and participates in polarization of ridges [36, 47, 48]. Conversely, as described above, early ommatidial polarization is mediated by Pk$^{sple}$, while later, Pk$^{pk}$ appears to mediate the middle and end of rotation. We investigated how the transition between isoforms occurs in the eye.

At the apico-lateral junctions of ommatidia, where PCP signaling occurs, microscopy shows that Pk$^{sple}$ is enriched in early ommatidia and Pk$^{pk}$ in late ommatidia. PCP-dependent rotation begins before this transition, and continues uninterrupted until finished well after the transition is complete (S5B Fig). Endogenously tagged isoforms show that apico-lateral Pk$^{sple}$ declines beginning at row 8 and is gradually replaced by Pk$^{pk}$ (Fig 2). Staining with the Pk[C] common antibody shows a brief period during the transition when the total signal is somewhat lowered (Fig 2). In a $pk^{pk}$ mutant, after a similar brief decline, Pk$^{sple}$ rather than declining further, increases to fully populate the apico-lateral junctions. In the wildtype, a potential explanation for the 'takeover' of apico-lateral localization by Pk$^{pk}$ is that the rise of Pk$^{pk}$ protein level induces a decline in Pk$^{sple}$ protein level. Imaging is not effective for evaluating the total cellular protein pools, so we probed whether protein level expression of one isoform is sensitive to expression of the other by performing Western blots of wildtype, $pk^{pk}$ and $pk^{sple}$ eye discs and pupal wings (Fig 7). Levels of Pk$^{pk}$ and Pk$^{sple}$ were not sensitive to presence or absence of the other in either tissue. Similarly, overexpression of $pk^{sple}$ did not alter the Pk$^{pk}$ protein level in eyes (Fig 7A). We therefore propose that the takeover from apico-lateral Pk$^{sple}$ to apico-lateral Pk$^{pk}$ is mediated by a competition for participation in the PCP signaling process at the apico-lateral junctions. We propose that increasing levels of Pk$^{pk}$ and a level of Pk$^{sple}$ that declines transiently facilitate the takeover. Supporting the idea that rising levels of Pk$^{pk}$ outcompete Pk$^{sple}$ for participation in PCP, an EGFP::Pk$^{pk}$ overexpressing clone reduced apico-lateral Pk$^{sple}$ signal in early ommatidia (10 independent clones in 8 eyes showed similar effects) (Fig 6C). Prior work has suggested that the kinase Nemo limits Pk$^{pk}$ protein levels [49], raising the possibility that decline of Nemo activity that may initially be high in early ommatidia could in part control a rise in Pk$^{pk}$ levels as ommatidia mature.

In the wing, prior data suggests the dichotomous possibilities that each Pk isoform can either sequester the other or outcompete the other for apico-lateral junctional localization [30, 31, 36]. We hypothesize that these outcomes might depend not only on levels but also on timing of expression. To test this idea, we used an assay in the wing in which we could recognize whether ectopically expressed Pk$^{sple}$ is either sequestered with Pk$^{pk}$ at its normal proximal location or if it out-competes Pk$^{pk}$ and reorients PCP signaling to the Ft/Ds/Fj signal. We found that immediately after pulsed expression, ectopic Pk$^{sple}$ is sequestered proximally by endogenous Pk$^{pk}$ (S9A and S9B Fig), but after additional time to accumulate, ectopic Pk$^{sple}$ out-competes endogenous Pk$^{pk}$ and reorients PCP (S9C and S9D Fig). By analogy, if a similar

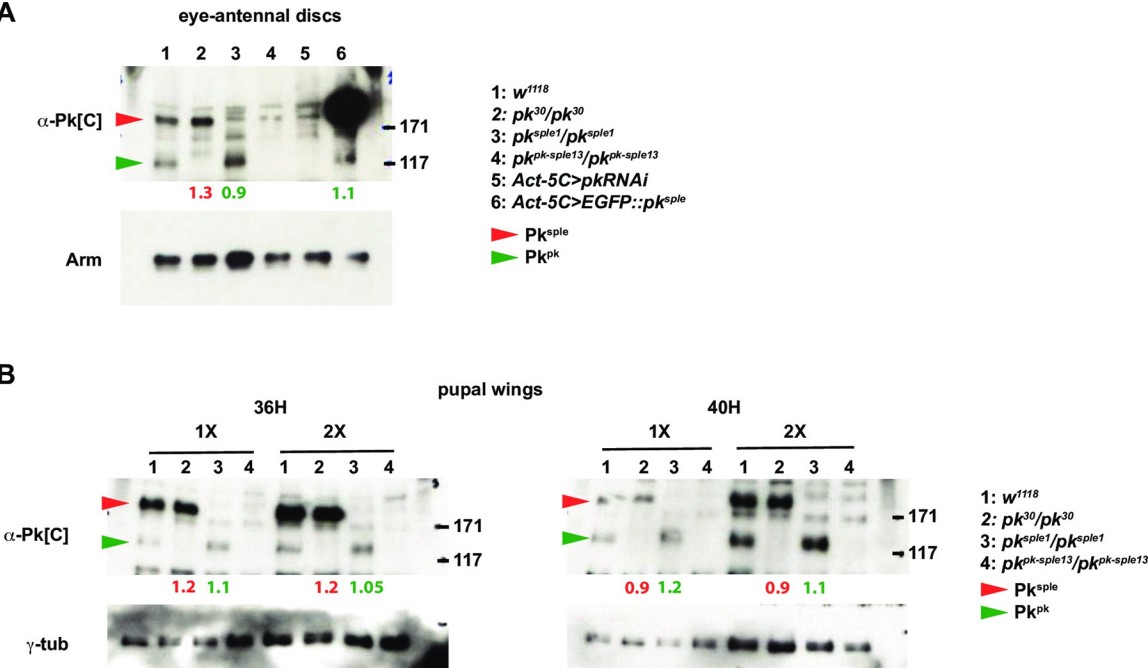

**Fig 7. Pk^pk and Pk^sple isoform protein levels are independent of the other in eye-antennal and pupal wing discs.** A. Western blot from eye-antennal discs probed with the common Pk[C] antibody detects Pk^pk and Pk^sple isoforms. Levels do not vary significantly in the presence or absence of the other, nor does Pk^pk vary upon overexpression of EGFP::Pk^sple. B. Western blots from 36H and 40H pupal wings probed with the common Pk[C] antibody detects Pk^pk and Pk^sple isoforms. At these times, declining Pk^pk levels and increasing Pk^sple levels result in the presence of both isoforms. Neither changes significantly in the absence of the other. 1X lysate loadings were doubled (2X) at each time point. Fold differences in levels relative to $w^{1118}$ (lanes 1), normalized to loading controls, are shown below the lanes for Pk^sple (red) and Pk^pk (green).

competition occurs in the eye, increasing Pk^pk expression is expected to first join Pk^sple and then gradually replace it in PCP signaling complexes, allowing rotation to continue uninterrupted. Note, however, that in the eye, no reorientation is expected since the Ft/Ds/Fj system does not contribute to rotation.

## Discussion

By dissecting the contributions of the Pk isoforms Pk^pk and Pk^sple to polarization of ommatidia, we define distinct PCP dependent polarization events: (i) biasing of the N-dependent R3/R4 differentiation, and (ii) facilitating rotation through 90˚. Differential expression dictates that the R3/R4 fate decision depends on Pk^sple, while rotation depends first on Pk^sple until that role is taken over by Pk^pk (Fig 8).

In addition to these two events, we consider the relationship between the R3/R4 cell fate decision and the direction of rotation. In wildtype ommatidia, the direction of rotation is intimately correlated with the R3/R4 fate decision, such that rotation always occurs in the direction of the R4 cell. Rotations are therefore in opposite directions in the dorsal and ventral fields. Even when R3:R4 fates are intermixed, such as in *ft*, *ds* or *fj* mutants, or in *pk^sple* mutant eyes [22, 38], this coordination is maintained so that ommatidia with dorsal or ventral morphology, whether located in the dorsal or ventral field, still rotate toward R4 (Fig 1). However, in other core mutants such as *pk^pk-sple*, *fz* and *vang*, many ommatidia of determinate chirality lack coordination between the direction of rotation and the R3/R4 fate choice, rotating toward

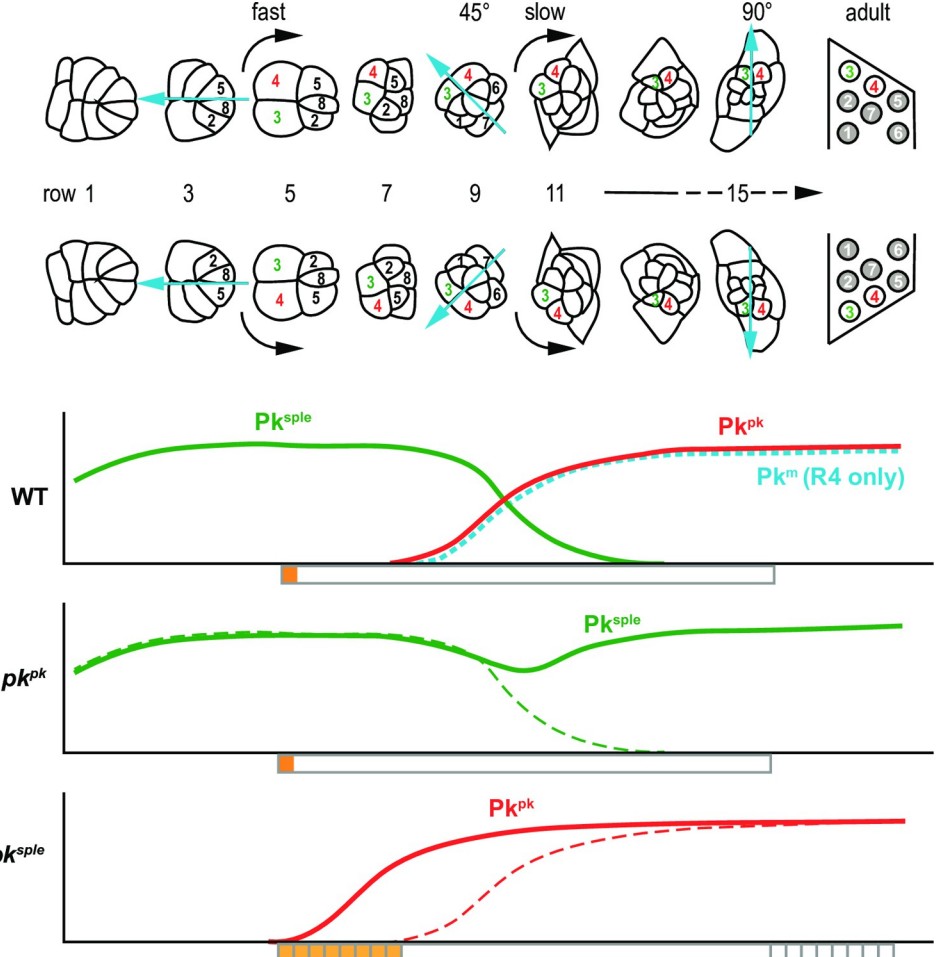

**Fig 8. Schematic of Pk isoform apico-lateral junctional localization and timing of polarization events.** Relative levels of apico-lateral junctional Pk$^{pk}$, Pk$^{sple}$ and Pk$^m$ isoforms, representing participation in PCP signaling, are graphed in relation to the timing of ommatidial maturation in wildtype (WT) and in $pk^{pk}$ and $pk^{sple}$ mutants. Orange and yellow bars below the graphs indicate the timing of the R3/R4 cell fate decision, and gray boxes indicate the timing of rotation. In $pk^{sple}$ mutants, R3/R4 decisions are delayed in some ommatidia as is the timing of rotation. Early Pk$^{sple}$-mediated decisions are responsive to Ft/Ds/Fj (orange) and whereas Pk$^{pk}$-mediated decisions are not (yellow).

R3 rather than R4 [22, 26]. Core PCP signaling is therefore required to coordinate these two events. Coordination is maintained in *ft*, *ds* and *fj* mutants, indicating that the Ft/Ds/Fj signal is not required.

Rotation begins around row 5 (S5 Fig), at approximately the same time the first known marker of R4 fate is expressed (the N reporter construct mδ0.5; [23]), but because cell fate might be determined slightly before this expression is evident, and because detecting the earliest rotation is imprecise, we cannot distinguish whether fate or rotation is established before the other or concurrently. Coordination might therefore occur by PCP signaling determining one event and that event triggering the other, or both events responding independently to the PCP signal.

Because coordination occurs when Pk$^{sple}$ expression is predominant, and because $pk^{pk}$ mutants are phenotypically normal, it is likely normally dependent on Pk$^{sple}$. Coordination nonetheless occurs normally in $pk^{sple}$ mutants. As discussed previously, apico-lateral junctional

Pk$^{pk}$ (or less likely, Pk$^m$) accumulation is detected at the time rotation begins in $pk^{sple}$ mutants (Fig 4), indicating that Pk$^{pk}$ can equally well mediate the coordination response, though with a brief delay in some ommatidia reflected in modest delays of both fate determination and rotation (S5 Fig). We therefore propose that coordinating rotation direction to the R3/R4 fate decision is a third distinct polarization decision.

At the time of the R3/R4 fate choice, Pk$^{sple}$ achieves an asymmetric distribution, accumulating on the equatorial sides of R3 and R4 where Vang accumulates and opposite Fz and Dsh on the polar sides of these cells (Fig 3). While the precise mechanism by which the core PCP system interacts with N is not known, the asymmetric localization strongly suggests that this event shares with hairs and bristles the feature that their distributions determine the polarity of downstream effectors.

The R3/R4 fate decision is made by a N-dependent competition between the two bipotent R3/R4 precursors. The N competition is imprecise, and in the absence of a PCP contribution, some R3/R4 pairs fail to decide, resulting in symmetric ommatidia, and the ones that decide do so with random orientation with respect to the equatorial-polar axis. Though the Pk$^{sple}$ isoform both facilitates efficient N signaling and biases the orientation direction, in its absence, biasing the orientation direction fails without compromising the efficiency of the N-dependent decision. Preservation of efficient R3/R4 fate determination in the $pk^{sple}$ mutant results sufficiently from early apico-lateral junctional localization of another isoform detected with the Pk [C] common antibody that we believe to be the Pk$^{pk}$ isoform rather than the Pk$^m$ isoform (Figs 4 and 8). We consider this is likely because, in wildtype, endogenously tagged Pk$^{pk}$ is expressed in most ommatidial cells beginning at approximately this time, whereas endogenously tagged Pk$^m$ expression is limited to the R4 cell (S7 and S8 Figs). Furthermore, we know that Pk$^{pk}$ can fulfill this function as it was previously shown that exogenously expressed Pk$^{pk}$ can rescue efficient R3/R4 fate decisions in a $pk^{sple}$ mutant [43]. We propose that absence of competition by Pk$^{sple}$ results in an earlier onset of apico-lateral junctional accumulation of Pk$^{pk}$ in the $pk^{sple}$ mutant relative to wildtype.

While in a $pk^{sple}$ mutant the rising expression and apico-lateral junctional localization of (presumed) Pk$^{pk}$ can activate PCP signaling to effectively bias N decisions, it does not rescue the random orientation of those decisions. Later and weaker Pk$^{pk}$ expression compared to Pk$^{sple}$ expression might account for this inability, but it was previously observed that *ActP-EGFP::pk$^{pk}$* expression that is present throughout eye development and is stronger than endogenous levels was also unable to rescue orientation whereas *ActP-EGFP::pk$^{sple}$* rescued all patterning defects including orientation [43]. A caveat to this experiment is that *ActP-EGFP:: pk$^{pk}$* expression alone produced orientation defects, indicating a dominant interference with the core PCP system as has been observed for other core components [23, 26, 39, 45]. Nonetheless, the likely explanation is that both Pk$^{sple}$ and Pk$^{pk}$ are able to mediate core PCP signaling to promote the N decision but core signaling incorporating Pk$^{pk}$ is unresponsive to the Ft/ Ds/Fj cue that normally orients that decision with respect to the equatorial-polar axis. We hypothesize that Pk$^{sple}$ responds to the Ft/Ds/Fj directional cue through its previously demonstrated binding to Ds and Dachs, but that the alternate microtubule based mechanism for linking Pk$^{pk}$ to Ft/Ds/Fj observed in wing and abdomen does not function in the eye. Furthermore, it appears that the time window for the Pk$^{sple}$ orientation response to Ft/Ds/Fj is limited, as late restoration of PCP signaling rescues R3/R4 determination but not orientation.

PCP signaling is required for ommatidial rotation, as both indeterminate and determined ommatidia fail to properly undergo 90˚ of rotation in core PCP mutants. Rotation begins while Pk$^{sple}$ participates in core PCP signaling, and continues during the takeover by Pk$^{pk}$ (Fig 8). Furthermore, both Pk$^{pk}$ and Pk$^{sple}$ can equally mediate the core PCP function in rotation, since elimination of either one but not the other permits rotation to proceed correctly to

completion. In the absence of either, the void in apico-lateral junctional localization left by that absence is filled by the other. Thus, for the rotation process, $Pk^{pk}$ and $Pk^{sple}$ are interchangeable. Their interchangeability is consistent with the absence of a contribution from the Ft/Ds/Fj system to rotation, as the only known difference in their activities is their differential response to this signal.

The takeover of Pk function in core PCP signaling by $Pk^{pk}$ appears to reflect first the recruitment of $Pk^{pk}$ into core complexes primarily occupied by $Pk^{sple}$, and may involve the known ability of the two isoforms to hetero-oligomerize [30]. As $Pk^{pk}$ levels progressively rise, $Pk^{pk}$ seems to outcompete and displace $Pk^{sple}$ from core complexes, such that $Pk^{sple}$ is no longer detectable in apico-lateral junctions of ommatidia, but likely persists in intracellular pools. We hypothesize that the takeover is aided by a transient decrease in apico-lateral $Pk^{sple}$ occupancy that temporally coincides with the second mitotic wave and the rising $Pk^{pk}$ levels, perhaps accelerating the ability of $Pk^{pk}$ to displace $Pk^{sple}$. The first 45° of rotation occur while under $Pk^{sple}$ control and proceed rapidly, whereas the second 45° of rotation occur under $Pk^{pk}$ control and proceed more slowly, but the interchangeability of isoforms suggests there is no causal relationship between isoform and speed of rotation.

The mechanism by which rotation occurs, and thus how PCP regulates rotation, is entirely unknown. Junctional remodeling between ommatidial cells and surrounding cells occurs in a stereotyped way, and in PCP mutants occurs variably, indicating that PCP signaling somehow steers rotation but does not itself propel it. One speculative hypothesis is that ommatidial cells produce motile processes that drive remodeling and that these processes need to be directionally coordinated, much like how PCP orients motile processes in socket cells to signal directionally in the induction of a bract [50], or how PCP is thought to direct motility of migrating cells [51]. Curiously, like bract induction, rotation requires N function in a mechanism that regulates EGF signaling [52].

In the course of examining isoform expression, we found that $Pk^m$ becomes the predominant isoform in R4 cells later in rotation. $Pk^m$ expression was previously thought to be limited to the embryo [29]. $pk^m$ mutants showed no apparent eye phenotype, and we have not created double mutants to probe for potential redundancy, so we cannot presently assign a function to this expression.

Coordination of R3/R4 fate determination with rotation direction normally takes place when $Pk^{sple}$ is predominant, but like rotation, it is maintained when either Pk isoform is present without the other. In $pk^{sple}$ mutants, the accelerated arrival of $Pk^{pk}$ at apico-lateral junctions, and the modest delay in ommatidial maturation are apparently sufficient to fulfill this role. As we observed for the rotation event itself, the interchangeability of the two isoforms in coordination is consistent with the absence of a Ft/Ds/Fj contribution and therefore the possibility of the differential response associated with $Pk^{pk}$ and $Pk^{sple}$ in other contexts.

It is intuitively appealing to hypothesize that rotation direction is a direct consequence of the orientation of the R3:R4 outcome, but models in which rotation direction or the R3:R4 outcome directly depend on the other make it difficult to understand how these events can lose coordination in core PCP mutants. Further insight will likely depend on gaining some understanding of the rotation mechanism.

We have revealed the molecular underpinning for the observation that the Pk isoforms $Pk^{pk}$ and $Pk^{sple}$ participate differentially in regulating polarization of the *Drosophila* eye, echoing similar findings for their function in the wing and bristles. In addition, we unexpectedly identified expression of the third isoform, $Pk^m$, in the eye, though our analyses did not identify a function. However, we also observed abundant expression in the larval brain, suggesting new avenues for exploration.

# Materials and methods

Key resources table

| Reagent type (species) or resource | Designation | Source or reference | Identifiers | Additional information |
|---|---|---|---|---|
| Genetic reagent (Drosophila melanogaster) | $pk^{pk\text{-}sple13}$ | **Gubb et al., 1999**, PMID: 10485852 | BDSC:41790; FLYB:FBal0060943; RRID:BDSC_41790 | FlyBase symbol: $pk^{pk\text{-}sple\text{-}13}$ |
| Genetic reagent (Drosophila melanogaster) | $pk^{pk\text{-}sple14}$ | **Gubb et al., 1999**, PMID: 10485852 | FLYB:FBal0035401 | FlyBase symbol: $pk^{pk\text{-}sple\text{-}14}$ |
| Genetic reagent (Drosophila melanogaster) | $pk^{pk30}$ | **Gubb et al., 1999**, PMID: 10485852 | BDSC:44229; FLYB:FBal0101223; RRID:BDSC_44229 | FlyBase symbol: $pk^{30}$ |
| Genetic reagent (Drosophila melanogaster) | $pk^{sple1}$ | **Gubb et al., 1999**, PMID: 10485852 | BDSC:422; FLYB:FBal0016024; RRID: BDSC_422 | FlyBase symbol: $pk^{sple\text{-}1}$ |
| Genetic reagent (Drosophila melanogaster) | $vang^{A3}$ | **Taylor et al., 1998**, PMID: 9725839 | FLYB:FBal0093183 | FlyBase symbol: $Vang^{A3}$ |
| Genetic reagent (Drosophila melanogaster) | $vang^{stbm6}$ | **Wolf and Rubin, 1998**, PMID: 9463361 | BDSC:6918; FLYB:FBal0062424; RRID:BDSC_6918 | FlyBase symbol: $Vang^{stbm\text{-}6}$ |
| Genetic reagent (Drosophila melanogaster) | $fz^{R52}$ | **Krasnow and Adler, 1994**, PMID:7924994 | FLYB:FBal0004939 | FlyBase symbol: $fz^{23}$ |
| Genetic reagent (Drosophila melanogaster) | $UAS\text{-} EGFP{::}pk^{sple}$ | **Gubb et al., 1999**, PMID: 10485852 | N/A | |
| Genetic reagent (Drosophila melanogaster) | $UAS\text{-}pk^{RNAi}$ | Vienna Drosophila Resource Center | VDRC:v101480; FLYB:FBst0473353; RRID:FlyBase_FBst0473353 | FlyBase symbol: $P\{KK109294\}$ [30–33, 53]VIE-260B |
| Genetic reagent (Drosophila melanogaster) | $UAS\text{-} EGFP{::}pk^{pk}$ | **Ehaideb et al., 2014**, PMID: 25024231 | FLYB:FBtp0112766 | Flybase symbol: $P\{UAS\text{-}eGFP.pk\}$ |
| Genetic reagent (Drosophila melanogaster) | $UAS\text{-}vang$ | **Lee et al., 2003**, PMID: 14562058 | FLYB:FBtp0020667; | Flybase symbol:$P\{UAS\text{-}stbm.L\}$ |
| Genetic reagent (Drosophila melanogaster) | $UAS\text{-}w^{RNAi}$ | B. Lu, Stanford, Stanford, USA | FLYB:FBal0282569 | Flybase symbol: $w^{dsRNA.UAS.cHa}$ |
| Genetic reagent (Drosophila melanogaster) | $V5{::}3Xmyc{::}APEX2{::}pk^{pk}$ | **Cho B et al., 2020**, PMID: 31934858 | FLYB:FBti0211290 | Flybase symbol: $TI\{TI\}pk^{pk\text{-}V5}$ |
| Genetic reagent (Drosophila melanogaster) | $V5{::}3Xmyc{::}APEX2{::} pk^{sple}$ | **Cho B et al., 2020**, PMID: 31934858 | FLYB:FBti0211291 | Flybase symbol: $TI\{TI\}pk^{sple\text{-}V5}$ |
| Genetic reagent (Drosophila melanogaster) | GMR-GAL4 | Bloomington Drosophila Stock Center | BDSC:84247; FLYB:FBti0210049; RRID:BDSC_84247 | Flybase symbol: $P\{GMR\text{-}GAL4.Y\}YH3$ |
| Genetic reagent (Drosophila melanogaster) | Act5C-GAL4 | Bloomington Drosophila Stock Center | BDSC:3954; FLYB:FBti0012292; RRID: BDSC_3954 | Flybase symbol: $P\{Act5C\text{-}GAL4\}$ 17bFO1 |
| Genetic reagent (Drosophila melanogaster) | armP-fz::EGFP | **Strutt, 2001**, PMID:11239465 | FLYB:FBtp0014592 | Flybase symbol: $P\{arm\text{-}fz.GFP\}$ |

(*Continued*)

**Fig 8.** (Continued)

| | | | | |
|---|---|---|---|---|
| Genetic reagent (Drosophila melanogaster) | *UAS-mCherry* | Bloomington Drosophila Stock Center | BDSC:38424; FLYB:FBti0147460; RRID:BDSC_38424 | Flybase symbol: *P{UAS-mCherry.NLS}3* |
| Genetic reagent (Drosophila melanogaster) | *actP>CD2>Gal4* | Bloomington Drosophila Stock Center | BDSC:30558; FLYB:FBti0012408; RRID:BDSC_30558 | Flybase symbol: *P{GAL4-Act5C(FRT.CD2).P}S* |
| Genetic reagent (Drosophila melanogaster) | *UAS-RFP* | Bloomington Drosophila Stock Center | BDSC:30558; FLYB:FBti0129814; RRID:BDSC_30558 | Flybase symbol: *P{UAS-RFP.W}3* |
| Genetic reagent (Drosophila melanogaster) | *tubP-GAL80$^{ts}$* | Bloomington Drosophila Stock Center | BDSC:7017; FLYB:FBti0027797; RRID: BDSC_7017 | Flybase symbol: *P{tubP-GAL80ts}2* |
| Genetic reagent (Drosophila melanogaster) | *pk$^{pk30}$ V5::3Xmyc:: APEX2::pk$^{sple}$* | In this study | N/A | |
| Genetic reagent (Drosophila melanogaster) | *pk$^{mCRISPR}$* | In this study | N/A | |
| Genetic reagent (Drosophila melanogaster) | *pk$^{pk-spleCRISPR}$* | In this study | N/A | |
| Genetic reagent (Drosophila melanogaster) | *HA::Pk$^m$* | In this study | N/A | |
| Antibody | Rat monoclonal anti-HA | Roche | Roche:11867423001 | 1/200 (Immunolabelling) |
| | | | RRID: AB_390918 | 1/1000 (Western blotting) |
| Antibody | Rat polyclonal anti-Pk$^{sple}$ | In this study | N/A | 1/200 (Immunolabelling) |
| Antibody | Mouse monoclonal anti-Fmi | DSHB | RRID:AB_528247 | 1/200 (Immunolabelling) |
| Antibody | Mouse monoclonal anti-V5 | Thermo-Fisher | Thermo_Fisher:R960-25, RRID:AB_2556564 | 1/200 (Immunolabelling) |
| | | | | 1/1000 (Western blotting) |
| Antibody | Guinea pig polyclonal anti-Pk[C] | ***Olofsson et al., 2014***, PMID:25005476 | N/A | 1/800 (Immunolabelling) |
| | | | | 1/1000 (Western blotting) |
| Antibody | Rabbit polyclonal anti-Vang | ***Rawls AS and Wolff T, 2003***, PMID: 12642492 | N/A | 1/400 (Immunolabelling) |
| Antibody | Rat monoclonal anti-dEcad | DSHB | RRID:AB_528120 | 1/200 (immunolabelling) |
| Antibody | Mouse monoclonal anti-Arm | DSHB | RRID: AB_528089 | 1/000 (Western blottinjg) |
| Antibody | Mouse monoclonal anti-γ-Tubulin | Sigma-Aldrich | Sigma-Aldrich: T6557 | 1/1000 (Western blotting) |
| | | | RRID:AB_477584 | |
| Recombinant DNA reagent | pCFD4 | Addgene | RRID:Addgene_49411 | CRISPR gRNA backbone |
| Recombinant DNA reagent | pDsRedattp | Addgene | RRID:Addgene_51019 | Donor recombinant DNA backbone |
| Sequence-based reagent | *pk* gRNA 1 | This paper | gRNA sequence in PCR primers | TTAGCGGAGTTCGGCTGAT |
| Sequence-based reagent | *pk* gRNA 2 | This paper | gRNA sequence in PCR primers | CGATCGGAAGAGGAAGCCC |
| Sequence-based reagent | *Pk$^m$* gRNA 1 | This paper | gRNA sequence in PCR primers | GTTTTCCTCAATCTCTTCGC |
| Sequence-based reagent | *Pk$^m$* gRNA 2 | This paper | gRNA sequence in PCR primers | CAGTGTTGCATGCAGCATAC |

## Fly strains and genetics

*Drosophila melanogaster* flies were grown on standard cornmeal/agar/molasses media at 25˚C. FLP-on (using the *actP>CD2>GAL4* construct for trans-gene expression) clones were generated by incubating second or third-instar larvae at 37˚C for 1 hr. 36 to 48 hrs later, discs were isolated from third instar larvae for staining and white prepupae were collected and aged to the desired developmental time point prior to dissection and fixation. Transient clonal expression of EGFP::Pk$^{sple}$ was induced by inactivating temperature sensitive GAL80 (GAL80$^{ts}$) in 30˚C for the indicated time period; GAL80$^{ts}$ lines were incubated in 18˚C.

The following alleles and stocks were used. FlyBase and VDRC ID numbers, when available, are in parentheses. Detailed chromosomes and genotypes are provided below. $pk^{pk\text{-}sple13}$ (FBal0060943), $pk^{pk\text{-}sple14}$ (FBal0035401) [44, 46], $pk^{pk30}$ (FBal0101223), $pk^{sple1}$ (FBal0016024), $vang^{A3}$ (FBal0093183) [49, 51, 54], $vang^{stbm6}$ (FBal0062423), $fz^{R52}$ (FBal0004939) [50, 52, 55], *UAS-EGFP::pk$^{sple}$* [29], *UAS-EGFP::pk$^{pk}$* (FBtp0112766), *UAS-vang* (FBtp0020667), *UAS-w$^{RNAi}$* (FBal0282569), *V5::3Xmyc::APEX2::pk$^{pk}$* (FBti0211290), *V5::3Xmyc::APEX2::pk$^{sple}$* (FBti0211291), $pk^{pk30}$ *V5::3Xmyc::APEX2::pk$^{sple}$* (in this study), $pk^{mCRISPR}$ (in this study), $pk^{pk\text{-}spleCRISPR}$ (in this study), *HA::Pk$^m$* (in this study), *UAS-pk$^{RNAi}$* (VDRC ID: 101480), *armP-fz::EGFP* (FBtp0014592) [51, 53, 55, 56], *UAS-mCherry* (FBti0147460), *actP>CD2>Gal4 UAS-RFP* (from B. Lu, Stanford, Stanford, USA), *tubP-Gal80$^{ts}$* (FBti0027797), *Act5C-GAL4* (FBti0012292), *GMR-GAL4* (FBti0210049).

## Genotypes of experimental models

**Fig 2.** (A) *V5::3Xmyc::APEX2::pk$^{sple}$/ V5::3Xmyc::APEX2::pk$^{sple}$*. (B) *V5::3Xmyc::APEX2::pk$^{pk}$/ V5::3Xmyc::APEX2::pk$^{pk}$*.

**Fig 3.** (A) $pk^{pk\text{-}sple13}$/$pk^{pk\text{-}sple14}$; *armP-fz::EGFP/ armP-fz::EGFP*. (B) *V5::3Xmyc::APEX2:: pk$^{sple}$/ V5::3Xmyc::APEX2::pk$^{sple}$*.

**Fig 4.** (A) $pk^{sple1}$/ $pk^{sple1}$. (B) $w^{1118}$. (C) $pk^{sple1}$/ $pk^{sple1}$.

**Fig 5.** $vang^{A3}$/ $vang^{stbm6}$; *UAS-w$^{RNAi}$/ GMR-GAL4*. $vang^{A3}$/ $vang^{stbm6}$; *UAS-vang/ GMR-GAL4*. *UAS-pk$^{RNAi}$/ gmr-GAL4*. $pk^{pk30}$/ $pk^{pk30}$; *UAS-pk$^{RNAi}$/ GMR-GAL4*. $pk^{sple1}$/ $pk^{sple1}$; *UAS-pk$^{RNAi}$/ GMR-GAL4*.

**Fig 6.** (A) $pk^{pk30}$/ $pk^{pk30}$. (B) $pk^{pk30}$ *V5::3Xmyc::APEX2::pk$^{sple}$/ $pk^{pk30}$ V5::3Xmyc::APEX2:: pk$^{sple}$*. (C) *y w hsflp/ +; V5::3Xmyc::APEX2::pk$^{sple}$/ V5::3Xmyc::APEX2::pk$^{sple}$; actP>CD2>GAL4 UAS-RFP/ UAS-EGFP::pk$^{pk}$*.

**Fig 7.** (A) lane 1: $w^{1118}$, lane 2: $pk^{pk30}$/ $pk^{pk30}$, lane 3: $pk^{sple1}$/ $pk^{sple1}$, lane 4: $pk^{pk\text{-}sple13}$/ $pk^{pk\text{-}sple13}$; lane 5: *UAS-pk$^{RNAi}$/ Act5C-GAL4*, lane 6: *UAS-EGFP::pk$^{sple}$/ Act5C-GAL4*. (B) lane 1: $w^{1118}$, lane 2: $pk^{pk30}$/ $pk^{pk30}$, lane 3: $pk^{sple1}$/ $pk^{sple1}$, lane 4: $pk^{pk\text{-}sple13}$/ $pk^{pk\text{-}sple13}$.

**S2 Fig** (A) *V5::3Xmyc::APEX2::pk$^{sple}$/ V5::3Xmyc::APEX2::pk$^{sple}$*. (B) *V5::3Xmyc::APEX2:: pk$^{pk}$/ V5::3Xmyc::APEX2::pk$^{pk}$*. (C) $w^{1118}$.

**S3 Fig** $w^{1118}$.

**S4 Fig** *V5::3Xmyc::APEX2::pk$^{pk}$/ V5::3Xmyc::APEX2::pk$^{pk}$*. **S5 Fig** (A, C) $pk^{sple1}$/ $pk^{sple1}$. (B) $w^{1118}$.

**S6 Fig** (A) *UAS-w$^{RNAi}$/ GMR-GAL4*. (B) *UAS-pk$^{RNAi}$/ GMR-GAL4*.

**S7 Fig** (A) $w^{1118}$. (B) *V5::3Xmyc::APEX2::pk$^{sple}$/ V5::3Xmyc::APEX2::pk$^{sple}$*. (C) *V5::3Xmyc:: APEX2::pk$^{pk}$/ V5::3Xmyc::APEX2::pk$^{pk}$*. (D) *HA::pk$^m$/ V5::3Xmyc::APEX2::pk$^{pk}$*.

**S8 Fig** (A) $pk^{mCRISPR}$/ $pk^{mCRISPR}$ (#2). (B) lane 1: $w^{1118}$, lane 2: $pk^{pk\text{-}spleCRISPR}$/ $pk^{pk\text{-}spleCRISPR}$ (#2), lane 3: $pk^{pk\text{-}spleCRISPR}$/ $pk^{pk\text{-}spleCRISPR}$ (#5), lane 4: $w^{1118}$, lane 5: $pk^{mCRISPR}$/ $pk^{mCRISPR}$ (#2), lane 6: $pk^{mCRISPR}$/ $pk^{mCRISPR}$ (#7), lane 7: $pk^{pk\text{-}spleCRISPR}$/ $pk^{pk\text{-}spleCRISPR}$ (#2), lane 8: $pk^{pk\text{-}spleCRISPR}$/ $pk^{pk\text{-}spleCRISPR}$ (#5), lane 9: $pk^{pk\text{-}sple13}$/ $pk^{pk\text{-}sple13}$. (C) lane 1: $w^{1118}$, lane 2: *HA::pk$^m$/ HA:: pk$^m$* (#1), lane 3: *HA::pk$^m$/ HA::pk$^m$* (#7). (D) $pk^{mCRISPR}$/ $pk^{mCRISPR}$ (#2).

S9 Fig (A-D) *actP>CD2>GAL4/ y w hsflp*; *V5::3Xmyc::APEX2::pk^{pk}/ V5::3Xmyc::APEX2::pk^{pk}*; *tubP-GAL80^{ts}/ UAS-EGFP::pk^{sple}*. (E) *y w hsflp/ +*; *V5::3Xmyc::APEX2::pk^{pk}/ V5::3Xmyc::APEX2::pk^{pk}*; *actP>CD2>GAL4 UAS-RFP/ UAS-EGFP::pk^{sple}*. (F) *y w hsflp/ +*; *V5::3Xmyc::APEX2::pk^{sple}/ V5::3Xmyc::APEX2::pk^{sple}*; *actP>CD2>GAL4 UAS-RFP/ UAS-EGFP::pk^{pk}*.

## Immunohistochemistry

Immunohistochemistry of third instar larval eye discs was performed as follows. Discs were fixed for 5–15 min in 4% paraformaldehyde in PBS at 4˚C. Fixed eye discs were washed two times in PBST (PBS with 0.1% Triton X-100). After blocking for 1 hr in 5% Bovine serum Albumin in PBST at 4˚C, discs were incubated with primary antibodies in the blocking solution overnight at 4˚C. Incubations with secondary antibodies were done for 90min at room temperature in PBST. Washes in PBST were performed three times after primary and secondary antibody incubation, and incubations in phalloidin (1:200 dilution), if required, were done in PBST for 15 min followed by wash at room temperature before mounting. Pupal wings were dissected at indicated developmental time points after puparium formation (apf). Pupae were removed from their pupal cases and fixed for 60–90 min in 4% paraformaldehyde in PBS at 4˚C. Wings were then dissected and extracted from the cuticle, and were washed two times in PBST. From blocking to staining, the same procedures were applied as used for disc staining. Stained samples were mounted in 15 μl Vectashield mounting medium (Vector Laboratories). Primary antibodies were as follows: mouse monoclonal anti-V5 (1:200 dilution, Thermofisher, R960-25), guinea pig polyclonal anti-Pk[C] (1:800 dilution, ([32]), mouse monoclonal anti-Fmi (1:200 dilution, DSHB), rat monoclonal anti-HA (clone 3F10, 1:200 dilution, Roche), rat monoclonal anti-dEcad (1:200 dilution, DSHB), rabbit polyclonal anti-Vang (1:400, Wolf T). To obtain a rat polyclonal anti-Pk^{sple} (1:200 dilution), Rat antiserum was raised against a Pk^{sple} specific N-terminal fragment that had been expressed as a 6×His fusion protein in E. coli and purified by using a Ni-NTA column (Qiagen). Antiserum production was performed by Josman Laboratories. Secondary antibodies from Thermo Fisher Scientific were as follows: 488-donkey anti-mouse, 488-goat anti-guinea pig, 488-goat anti-rabbit, 546-donkey anti-goat, 633-goat anti-guinea pig, 633-goat anti-rat, 647-donkey anti-mouse. Alexa 635 and Alexa 350 conjugated phalloidin were from Thermo Fisher Scientific.

## Imaging and quantification

Adult eye plastic sections were generated [57] and imaged on a Nikon Eclipse E1000M equipped with a Spot Flex camera (Model 15.2 64 MP). All immunofluorescence images were taken with a Leica TCS SP8 AOBS confocal microscope and processed with LAS X (Leica) and Adobe Photoshop.

## CRISPR/Cas9 homology directed recombination for generating *pk* mutants and for tagging endogenous *pk* isoforms

**Construction of gRNA containing plasmids.** The pCFD4 plasmid (Addgene, 49411) was used to express two gRNAs under the U6 promoter. Using Gibson Assembly (NEB), gRNA sequences to generate two cleavage sites in the *pk* genomic locus were assembled into pCFD4 digested by the BbsI restriction enzyme. gRNA sequences were as follows:

*pk* gRNA1 (5′-TTAGCGGAGTTCGGCTGAT-3′) and *pk* gRNA2 (5′-CGATCGGAAGAG-GAAGCCC-3′) for *pk^{pk-spleCRISPR}* null mutants;

*pk^m* gRNA1 (5′- GTTTTCCTCAATCTCTTCGC -3′) and *pk^m* gRNA2 (5′-CAGTGTTGCATGCAGCATAC -3′) for *pk^{mCRISPR}* mutant and HA tagged Pk^m.

Construction of tagged $pk^{pk}$ and $pk^{sple}$ was reported previously [36], and the same gRNA sequences used for generating $V5::3Xmyc::APEX2$ tagged $pk^{sple}$ were used to tag $pk^{sple}$ in a chromosome bearing $pk^{pk30}$.

Stable transgenic flies expressing two gRNAs were generated by BestGene using the PhiC31 standard injection method.

**Generation of mutants and tagged alleles using CRISPR HDR.**  To generate pk^{pk-sple} and pk^m null mutants, and to introduce the HA tag at the N-terminus of pk^m, the CRIPR HDR technique was applied. Briefly, pk^{pk-sple} and pk^m null mutant were generated by deleting the DNA sequences between the two gRNA cleavage sites (for pk^{pk-sple} null, deletion of five C-terminal common exons; for pk^m null, part of the 5'UTR, the first exon, and part of the first intron of the pk^m genomic region) and replacing them by DsRed sequence from the pDsRed-attp (Addgene, 51019) plasmid with two homology arms. To generate N-terminal HA tagged pk^m, the DNA sequence was replaced by the same pk^m genomic sequence bearing HA tag at the N terminus of pk^m using the donor pDsRed-attp plasmid with homology arms. The donor plasmids containing homology arms, or homology arms together with the HA tag sequence, were sequenced and then injected into the stable transgenic flies expressing two gRNAs and nosCas9 to generate recombinants. The same CRISPR HDR strategy [36] was applied to tag pk^{sple} in the pk^{pk30} mutant chromosome. DsRed signal in the fly eyes was monitored for selecting the recombinants, and dsRed and flanking sequences were removed by the Cre-Lox site-specific recombination method. The resulting modified alleles are referred to in the text as pk^{pk-spleCRISPR} and pk^{mCRISPR} for the pk^{pk-sple} and pk^m CRISPR null mutants, respectively, and HA::Pk^m for the HA tagged Pk^m.

## Western blots

Eye antenna discs and brains from third-instar larvae, and pupal wings at appropriate developmental stages, were dissected and lysed in protein loading buffer. Lysates from eight discs or wings, or two brains, were loaded per lane for SDS-PAGE analysis and western blots were performed using standard procedures. Antibodies: Guinea pig polyclonal anti-Pk[C] (1:1,000 dilution; the same antibody used for immunostaining), mouse monoclonal anti-γ-Tubulin (1:1,000 dilution, Sigma-Aldrich, T6557), rat monoclonal anti-HA (clone 3F10, 1:1,000 dilution; the same antibody used for immunostaining), mouse monoclonal anti-Arm (1:1,000, DSHB). Secondary antibodies were Peroxidase-conjugated goat anti-guinea pig (1:10,000), goat anti-mouse (1:10,000), goat anti-rat (1:10,000) antibodies (all from Jackson Immuno Research), and detection used SuperSignal West Pico Chemiluminescent Substrate (Thermo-Fisher, 34080)

## Contact for reagent and resource sharing

Further information and requests for resources and reagents should be directed to the corresponding author, Jeffrey D. Axelrod (jaxelrod@stanford.edu).

## Supporting information

**S1 Fig. Raw images of Western blots shown in other Figures.**
(PDF)

**S2 Fig.**  V5::Pk^{sple} (A) and V5::Pk^{pk} (B) eyes co-stained with the common Pk antibody Pk[C]. The common antibody detects signal in apico-lateral junctions of all ommatidia, whereas V5::Pk^{sple} reports signal selectively in apico-lateral junctions of young ommatidia and V5::Pk^{pk} reports signal selectively in apico-lateral junctions of older ommatidia. C. Control eye co-

stained with anti-Pk$^{sple}$ and Pk[C]. The anti-Pk$^{sple}$ antibody detects early ommatidia, similar to V5::Pk$^{sple}$.
(PDF)

**S3 Fig. Vang and Pk isoforms co-localize in ommatidia.** A control ($w^{1118}$) eye stained with common Pk[C] (red in merge), anti-Vang (green in merge) and anti-dECad (blue in merge) antibodies. Vang co-localizes with all Pk isoforms in the apico-lateral junctions of ommatidia.
(PDF)

**S4 Fig. Vang co-localizes with Pk$^{pk}$ isoform in older ommatidia.** A $V5::pk^{pk}$ eye stained with anti-V5 (red in merge), Vang (green in merge) and dECad (blue in merge) antibodies. Vang co-localizes with Pk$^{pk}$ in the apico-lateral junctions of older ommatidia.
(PDF)

**S5 Fig. Asymmetry and rotation in control and $pk^{sple}$ eyes.** A. A $pk^{sple}$ eye stained for Vang (green) and dECad (blue) showing asynchronous acquisition of molecular asymmetry. Two row 6 ommatidia are labeled, one showing Vang asymmetry (yellow) and the other showing symmetric distribution. B. A control ($w^{1118}$; B), and C. a $pk^{sple}$ mutant eye stained for dECad with approximate orientations of ommatidia shown with arrows. While rotation is approximately synchronous in the control eye (B), rotation is highly asynchronous in the $pk^{sple}$ mutant eye (C).
(PDF)

**S6 Fig. GMR-GAL4 knocks down expression beginning around row 8.** Control ($GMR>w^{RNAi}$) and $pk$ knockdown ($GMR>pk^{RNAi}$) eyes labeled with anti-Pk[C] (green in merge) and anti-Fmi (red in merge) antibodies show knockdown of $pk$ in older ommatidia in the $GMR>pk^{RNAi}$ eye.
(PDF)

**S7 Fig. Pk$^{m}$ in eyes.** A. The Pk common antibody (Pk[C]) detects signal in R4 of older ommatidia, recognizable by their intense Fmi staining. B-C. $V5::pk^{sple}$ and $V5::pk^{pk}$ eyes labeled for V5 and Pk[C] show absence of V5::Pk$^{sple}$ and V5::Pk$^{pk}$ singal in R4's of older ommatidia. V5::Pk$^{sple}$ is still weakly detected in R4 at row 9 (Ba) but is absent from R4 by row 11 (Bb). V5::Pk$^{pk}$ is not detected in row 11 (Ca) or row 13 (Cb) ommatidia. D. A $HA::pk^{m}$, $V5::pk^{pk}$ eye stained for HA, Pk[C] and V5. HA::Pk$^{m}$ is seen in older R4 cells and V5::Pk$^{pk}$ is detected in other cells but not in R4. The dotted region is enlarged on the right.
(PDF)

**S8 Fig. Pk$^{m}$ expression in eye and brain.** A. A $pk^{m}$ mutant eye stained with the Pk[C] common antibody and anti-Fmi shows that no Pk signal is detected in the R4 cells of late ommatidia (arrowheads). B. A Western blot from control eye and larval brain probed with the Pk[C] antibody detects little if any Pk$^{m}$ in eyes, but Pk$^{m}$ is the predominant isoform in brain. C. A Western blot from control and HA::Pk$^{m}$ larval brain probed sequentially with anti-HA and the Pk[C] common antibody shows detection of endogenous Pk$^{m}$ in control and HA-tagged Pk$^{m}$ in the tagged strains. The HA::Pk$^{m}$ #1 lane is slightly under-loaded relative to the other lanes, and uneven developing occurred in the anti-HA blot. Both HA::Pk$^{m}$ lines therefore express at similar levels to endogenous Pk$^{m}$. D. A section of a $pk^{m}$ mutant eye shows normal polarity.
(PDF)

**S9 Fig. Timing and expression level determines outcome of competition between Pk$^{pk}$ and Pk$^{sple}$.** A-D. Pulsed expression of Pk$^{sple}$ in clones in the wing, allowed to accumulate at the permissive temperature (30°) starting at the indicated times and allowed to accumulate or decay

for varying times. Heat shocks at 37˚ induced clones. Flies were maintained at the restrictive temperature (18˚) until expression was induced at 30˚ (blue lines). The expected protein accumulation is shown in red, and equivalent developmental time is shown at the top. Modest expression first shows that Pk$^{sple}$ accumulates on the proximal sides of cells where Pk$^{pk}$ is already present (A-B). After accumulating to a higher level, Pk$^{sple}$ takes over and polarity is reoriented with Pk$^{sple}$ localizing to the posterior as expected from the Ds/Fj gradients (C-D; clones in the posterior wing). E. High level tagged Pk$^{sple}$ clonal expression beginning in third instar in V5::Pk$^{pk}$ wings shows that high levels of Pk$^{sple}$ out-compete and displace Pk$^{pk}$ from apico-lateral junctions at 28H after puparium formation. F. High level tagged Pk$^{pk}$ clonal expression beginning in third instar in V5::Pk$^{sple}$ wings shows that high levels of Pk$^{pk}$ out-compete and displace Pk$^{sple}$ from apico-lateral junctions at 40H after puparium formation, when Pk$^{sple}$ is the predominant isoform [36]. 8 to 10 clones were analyzed for each condition and representative clones are shown.
(PDF)

## Acknowledgments

We thank Mike Simon and members of the Axelrod lab for useful discussions and critical reading of the manuscript.

## Author Contributions

**Conceptualization:** Bomsoo Cho, Jeffrey D. Axelrod.

**Funding acquisition:** Jeffrey D. Axelrod.

**Investigation:** Bomsoo Cho, Song Song, Joy Y. Wan.

**Writing – original draft:** Jeffrey D. Axelrod.

**Writing – review & editing:** Bomsoo Cho, Song Song, Joy Y. Wan, Jeffrey D. Axelrod.

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
