## [Decision Letter · Decision Letter 0]

26 Oct 2021

PONE-D-21-29540Prickle isoform participation in distinct polarization events in the Drosophila eyePLOS ONE

Dear Dr.  Axelrod,

Thank you for submitting your manuscript to PLOS ONE. After careful consideration, we feel that it has merit but does not fully meet PLOS ONE’s publication criteria as it currently stands. Therefore, we invite you to submit a revised version of the manuscript that addresses the points raised during the review process.

Please address all the reviewers’ comments, pay particular attention to include sample number and appropriate statistical analysis when necessary.

Please submit your revised manuscript within 4 weeks. If you will need more time than this to complete your revisions, please reply to this message or contact the journal office at plosone@plos.org. Please include the following items when submitting your revised manuscript:A rebuttal letter that responds to each point raised by the academic editor and reviewer(s). You should upload this letter as a separate file labeled 'Response to Reviewers'.A marked-up copy of your manuscript that highlights changes made to the original version. You should upload this as a separate file labeled 'Revised Manuscript with Track Changes'.An unmarked version of your revised paper without tracked changes. You should upload this as a separate file labeled 'Manuscript'.If applicable, we recommend that you deposit your laboratory protocols in protocols.io to enhance the reproducibility of your results. Protocols.io assigns your protocol its own identifier (DOI) so that it can be cited independently in the future. For instructions see: https://journals.plos.org/plosone/s/submission-guidelines#loc-laboratory-protocols. Additionally, PLOS ONE offers an option for publishing peer-reviewed Lab Protocol articles, which describe protocols hosted on protocols.io. Read more information on sharing protocols at https://plos.org/protocols?utm_medium=editorial-email&utm_source=authorletters&utm_campaign=protocols.

We look forward to receiving your revised manuscript.

Kind regards,

Carlos Oliva, PhD

Academic Editor

PLOS ONE

Journal Requirements:

PONE-D-21-29540

Reviewers' comments:

Reviewer's Responses to Questions

**Comments to the Author**

1. Is the manuscript technically sound, and do the data support the conclusions?

Reviewer #1: Yes

Reviewer #2: Yes

2. Has the statistical analysis been performed appropriately and rigorously? 

Reviewer #1: N/A

Reviewer #2: No

3. Have the authors made all data underlying the findings in their manuscript fully available?

Reviewer #1: Yes

Reviewer #2: Yes

4. Is the manuscript presented in an intelligible fashion and written in standard English?

Reviewer #1: Yes

Reviewer #2: Yes

5. Review Comments to the Author

Reviewer #1: The work presented in this manuscript helps to provide molecular explanation for previous genetic observations regarding the role of isoforms of the Prickle PCP protein in ommatidia polarity determination in the Drosophila eye. The data is generally well-presented although some images are not perfectly clear and some necessary introduction is lacking. The text seems a bit repetitive at times and would benefit from more thorough referencing whenever a specific point is made. The interpretations seem largely reasonable, although there a few bits that I’m not convinced about (noted below).

Specific points:

Generally, it would be good if most (all?) statements describing published results gave the appropriate reference. As the format is numbered references, this shouldn’t disrupt the flow of the text. I frequently found myself reading a statement and wondering what the evidence might be, but not knowing where to look. The Results section is particularly short on referencing.

“apical membrane” seems to be consistently used to mean “apico-lateral membrane/junctions”? Isn’t the “apical membrane” the membrane covering the apical part of the cell?

lines 7-9: The authors say that it is the protein expression analysis that reveals different functions of Pk[pk] and Pk[sple] isoforms, but really the key evidence is analysis of the mutant phenotypes (see also comment on lines 469-470). The results in this manuscript provide a molecular explanation for the previous phenotypic observations, but do not (cannot?) prove molecular function.

lines 85-87: Do we know that in PCP mutations rotation direction is uncoupled from R3:R4 orientation? Looking at the diagrams in Fig.1 it appears that the phenotypes could result from R3:R4 fate reversals and normal but >90° rotation direction relative to R3:R4 positions? I suppose the argument is that >90° rotation cannot occur? No reference is given for this specific statement (although I didn’t fully read refs 22-27 to check). However, I seem to recall that R3:R4 fates cannot be accurately scored by rhabdomere position in the adult eye, as in PCP mutants this can vary according to apico-basal depth, so the evidence would need to be from disc studies (and/or live imaging?).

lines 94-95: Is it really known that in “Pk[pk]-predominant tissues” core PCP is responding indirectly to the Ft/Ds/Fj signal? I recall quite a few papers showing fairly normal PCP in Ft/Ds/Fj mutant tissue, none of which are cited here (surprisingly). Doesn’t this suggest core PCP responding to upstream signals independently of Ft/Ds/Fj in some contexts? I have the feeling the authors are not providing a very balanced discussion, but I’m unsure why, as presenting a complete background doesn’t undermine their results (and supports their later conclusion that Pk[pk] alone is unable to couple to Ft/Ds/Fj).

Line 161/Fig.2B: the text says Pk[pk] is enriched at apical membranes (meaning apico-lateral junctions I assume, rather than the apical membrane of the cell?), but the predominant pattern seems to be blobs inside the cells, with maybe some hazy/weak membrane localization? Do the authors have a better image to make the point more convincingly? Even dEcad seems poor in this example (compare Pk[sple] row 7 and 9 images with Pk[pk] row 7 and 9 images).

Line 191: Pk[m] suddenly appears without having been previously mentioned? Some discussion in the Introduction (and some references) might help here?

Lines 188-195: If the signal detected by anti-Pk[C] is Pk[pk], and if “our experience suggests high sensitivity with the V5 antibody”, then the authors just need to look at V5::Pk[pk] in a pk[sple] background? Actually, I suspect they would need to engineer a sple lesion on top of V5::pk[pk] to do this (or sple specific RNAi?), which might be unattractive. However, if this is the case, maybe explain to the reader who may otherwise be puzzled, especially as the molecular biology of the pk, sple and m isoforms hasn’t been explained.

Fig.7: Pupal wing western blots are 36 hours and 40 hours APF at 25°C (36H and 40H)? So after hair polarity but during wing ridge polarity specification? Why these stages?

Line 337: Should be Fig.S7?

Line 338: I take the authors’ point, but is ectopic Pk[sple] “out-competing” Pk[pk] or just acting in parallel? I assume both accumulate in “proximal” complexes, but Pk[sple] is coupling to Ft/Ds/Fj to re-orient the polarity axis while Pk[pk] continues to act to reinforce cellular polarity? I suppose it might be “out-competing” if Pk[pk] was competing to orient polarity on the proximo-distal axis, but is there evidence for this?

Lines 469-470: Surely it’s the different phenotypes of pk[pk] vs pk[sple] vs pk[pk-sple] that reveal the different functions of the Pk[pk] and Pk[sple] isoforms in regulating PCP in the eye? The work in this manuscript provides some molecular explanation for those findings, but is not the key work demonstrating the different functions.

Fig.S6-2C: HA::Pk[m] #1 and #7 are knock-ins of HA that lead to vastly different levels of expression of HA::Pk[m]? This seems unexpected, so maybe the authors could add some words of explanation in the figure legend?

Reviewer #2: This very interesting article by Cho et al., describes Pk isoforms Pksple and Pkpk's expression and function during planar polarization of the Drosophila eye. The authors provide a detailed analysis of the expression patterns on Pk isoforms, including the undescribed Pkm isoform. The authors propose that the amount of Pkpk seen in Pk sple mutant would be more than enough to prevent mistakes in R3/R4 cell fate decisions although it can´t orient them along the dorsal-ventral axis. On the other hand both Pkpk and Pksple can sustain proper ommatidia rotation. Finally the authors propose that Pkpk competes with Pksple for localization at the apical membrane.

Major comment

Although the article is well structured and provides clarifying experiments regarding Pk function in the eye, it lacks replicates and statistical analyses. The authors do not mention whether the observations are made based on one case or different biological replicates. They do not provide statistical analysis or quantification of stainings and phenotypes. It could be understood to confirm well-described observations, but one expects new observations based on more quantitative data, for instance, quantification of immunostaining of Pk in the pksple background or Pksple in a pkpk background. Also, ommatidia maturation based on Vang staining.

I believe this is an essential part of the study, and it should be better characterized.

Similarly, phenotypic observations in the vang rescue experiment seem to be based only on one eye section, clonal analysis in one clone, and western blots do not appear to have replicates.

Minor comments

Calibrations bars are missing in all figures

legend figure 3: the authors claim that there is colocalization , however there is no colocalization analysis. Also they claim that the asymmetry increases, which is not clear from the picture. The authors should provide a quantitative analysis that supports their observations and indicate the asymmetry using arrows or arrowheads.

Line 185-186: Authors should indicate figure 2 when they mentioned that endogenously tagged Pkpk in wt ommatidia becomes highly expressed around row 9.

line 262: missing the number of the suppl figure (Fig 5 and S Fig)

6. PLOS authors have the option to publish the peer review history of their article (what does this mean?). If published, this will include your full peer review and any attached files.

Reviewer #1: No

Reviewer #2: No

---

## [Author Response · Author response to Decision Letter 0]

24 Nov 2021

We thank the reviewers for their careful reading and thoughtful comments. We have addressed them with requested modifications to the manuscript, and our point by point responses are indicated following the word “Response” below.

Reviewers' comments:

Reviewer's Responses to Questions

Comments to the Author

1. Is the manuscript technically sound, and do the data support the conclusions?

Reviewer #1: Yes

Reviewer #2: Yes

2. Has the statistical analysis been performed appropriately and rigorously?

Reviewer #1: N/A

Reviewer #2: No

3. Have the authors made all data underlying the findings in their manuscript fully available?

Reviewer #1: Yes

Reviewer #2: Yes

4. Is the manuscript presented in an intelligible fashion and written in standard English?

Reviewer #1: Yes

Reviewer #2: Yes

5. Review Comments to the Author

Reviewer #1: The work presented in this manuscript helps to provide molecular explanation for previous genetic observations regarding the role of isoforms of the Prickle PCP protein in ommatidia polarity determination in the Drosophila eye. The data is generally well-presented although some images are not perfectly clear and some necessary introduction is lacking. The text seems a bit repetitive at times and would benefit from more thorough referencing whenever a specific point is made. The interpretations seem largely reasonable, although there a few bits that I’m not convinced about (noted below).

Response: We eliminated some repetitive language in the introduction.

Specific points:

Generally, it would be good if most (all?) statements describing published results gave the appropriate reference. As the format is numbered references, this shouldn’t disrupt the flow of the text. I frequently found myself reading a statement and wondering what the evidence might be, but not knowing where to look. The Results section is particularly short on referencing.

Response: More thorough referencing, particularly in the results section, is now provided. 

“apical membrane” seems to be consistently used to mean “apico-lateral membrane/junctions”? Isn’t the “apical membrane” the membrane covering the apical part of the cell?

Response: The referee is correct, and we have changed to this more precise wording.

lines 7-9: The authors say that it is the protein expression analysis that reveals different functions of Pk[pk] and Pk[sple] isoforms, but really the key evidence is analysis of the mutant phenotypes (see also comment on lines 469-470). The results in this manuscript provide a molecular explanation for the previous phenotypic observations, but do not (cannot?) prove molecular function.

Response: We take the referees point, although in this passage we didn’t claim function. Here, we said the work shows distinct events with distinct contributions of the Pk isoforms. Nevertheless, we have altered the wording to reflect the spirit of the referee’s comment.

lines 85-87: Do we know that in PCP mutations rotation direction is uncoupled from R3:R4 orientation? Looking at the diagrams in Fig.1 it appears that the phenotypes could result from R3:R4 fate reversals and normal but >90° rotation direction relative to R3:R4 positions? I suppose the argument is that >90° rotation cannot occur? No reference is given for this specific statement (although I didn’t fully read refs 22-27 to check). However, I seem to recall that R3:R4 fates cannot be accurately scored by rhabdomere position in the adult eye, as in PCP mutants this can vary according to apico-basal depth, so the evidence would need to be from disc studies (and/or live imaging?).

Response: The referee is correct that any final angle can be gotten to by rotating in either direction if rotation of more than 90 degrees can occur. However, since the ommatidia in a single eye are effectively a time series, this would require that the younger ommatidia are rotating very fast to get to their final position by going the long way around, and this does not appear to be the case. 

lines 94-95: Is it really known that in “Pk[pk]-predominant tissues” core PCP is responding indirectly to the Ft/Ds/Fj signal? I recall quite a few papers showing fairly normal PCP in Ft/Ds/Fj mutant tissue, none of which are cited here (surprisingly). Doesn’t this suggest core PCP responding to upstream signals independently of Ft/Ds/Fj in some contexts? I have the feeling the authors are not providing a very balanced discussion, but I’m unsure why, as presenting a complete background doesn’t undermine their results (and supports their later conclusion that Pk[pk] alone is unable to couple to Ft/Ds/Fj).

Response: The referee is correct that there is evidence, particularly from the wing, that polarity is not severely impaired by some interventions that disrupt normal Ft/Ds/Fj function. This suggests the real possibility that other signals can provide input to core PCP signaling, but it says nothing to dispute the possibility that core PCP responds to Ft/Ds/Fj. It is critical to make the distinction between “responding to Ft/Ds/Fj” from “might also respond to other signals,” both of which are likely true. We think there’s very strong evidence that at least in wing, core PCP does respond to Ft/Ds/Fj [cf Fig 7 in Cho et al 2020 eLife] and a detailed mechanism for this has been proposed (references cited in the text). This is indeed a complex subject, but we think it would be a distraction to address it in any detail in this context

Line 161/Fig.2B: the text says Pk[pk] is enriched at apical membranes (meaning apico-lateral junctions I assume, rather than the apical membrane of the cell?), but the predominant pattern seems to be blobs inside the cells, with maybe some hazy/weak membrane localization? Do the authors have a better image to make the point more convincingly? Even dEcad seems poor in this example (compare Pk[sple] row 7 and 9 images with Pk[pk] row 7 and 9 images).

Response: The apical surfaces of early ommatidial cells (those closest to the morphogenetic furrow (mf)) are large enough to allow clear visualization of the apico-lateral junctions. The nuclei are still close to the apical surface in these Photoreceptor (PR) cells to cause the apical surface areas to be large. However, the nuclei in older ommatidia (further from the mf) migrate basally and the apical surfaces of these cells are too small to allow clear visualization of their apico-lateral junctions with conventional confocal microscopy, as is can be seen in the Ecad channel. Therefore, we cannot distinguish from this type of imaging whether signal is “blobs” inside cells or apico-lateral staining. Given that these proteins are seen predominantly at apico-lateral juncions in nearly all contexts where that can be visualized, we think it is reasonable to presume the same is true here. We do not specify which PR cells express Pkpk in later rows (from row 9 to 13) as the apices are too small identify individual cells. 

Line 191: Pk[m] suddenly appears without having been previously mentioned? Some discussion in the Introduction (and some references) might help here?

Response: We now introduce Pkm together with the other isoforms on lines 92-96.

Lines 188-195: If the signal detected by anti-Pk[C] is Pk[pk], and if “our experience suggests high sensitivity with the V5 antibody”, then the authors just need to look at V5::Pk[pk] in a pk[sple] background? Actually, I suspect they would need to engineer a sple lesion on top of V5::pk[pk] to do this (or sple specific RNAi?), which might be unattractive. However, if this is the case, maybe explain to the reader who may otherwise be puzzled, especially as the molecular biology of the pk, sple and m isoforms hasn’t been explained.

Response: We failed to generate viable homozygous flies after CRISPR engineering to add V5 to pkpk in the pksple mutant chromosome. There may have been off target effects or there may have been unexpected regulatory effects of the engineered modifications that resulted in homozygous lethality. This is now noted in the text lines 208-210.

Fig.7: Pupal wing western blots are 36 hours and 40 hours APF at 25°C (36H and 40H)? So after hair polarity but during wing ridge polarity specification? Why these stages?

Response: To measure the effect of one on the other, both isoforms need to be expressed. Enough Pksple together with Pkpk is expressed to be detected by western blots at these pupal stages. Before 32H apf, Pksple is expressed only in cells of the anterior wing margin, and this expression is insufficient to be detected with western blots.

Line 337: Should be Fig.S7?

Response: Corrected

Line 338: I take the authors’ point, but is ectopic Pk[sple] “out-competing” Pk[pk] or just acting in parallel? I assume both accumulate in “proximal” complexes, but Pk[sple] is coupling to Ft/Ds/Fj to re-orient the polarity axis while Pk[pk] continues to act to reinforce cellular polarity? I suppose it might be “out-competing” if Pk[pk] was competing to orient polarity on the proximo-distal axis, but is there evidence for this?

Response: We and others have shown that low levels of Pksple or Pkpk follow localization of the dominant Pk isoform in various tissues, apparently because of the ability of these proteins to oligomerize (Pkpk-Pkpk, Pkpk-Pksple or Pksple-Pksple) as well as their interaction with Vang. In our previous study in the wing and bristles, Pksple follows proximal localization of Pkpk, and we presume that Pksple at the proximal position is not directly coupled to the Ft/Ds global signal. In contrast, when the ratio of Pksple to Pkpk becomes higher, polarity is reversed, with Pksple localizing distally along with reversal in position of other core components and the morphological polarity outcome. We interpret this as Pksple functionally out-competing Pkpk to control the orientation of polarization. Pksple can only engage with Ds and D when it localizes distally, and its ability to interact with other components reverses the outcome when there is a sufficient amount of Pksple for these interactions to dominate.

Lines 469-470: Surely it’s the different phenotypes of pk[pk] vs pk[sple] vs pk[pk-sple] that reveal the different functions of the Pk[pk] and Pk[sple] isoforms in regulating PCP in the eye? The work in this manuscript provides some molecular explanation for those findings, but is not the key work demonstrating the different functions.

Response: We’ve modified the language to be more clear about the contribution of this work (now line 571).

Fig.S6-2C: HA::Pk[m] #1 and #7 are knock-ins of HA that lead to vastly different levels of expression of HA::Pk[m]? This seems unexpected, so maybe the authors could add some words of explanation in the figure legend?

Response: We believe this apparent difference is due to unequal loading and aberrant staining. The same blot was probed sequentially with anti-HA and anti-Pk[C]. As can be seen in the anti-Pk[C] staining, the HA::Pkm #1 lane is slightly underloaded relative to the others based on the relative intensities of the non-specific faster migrating band visible in the uncropped original blot. We think the apparently larger difference in the anti-HA stain is due to residual developing solution that overlaps the Pkm band in #7 and a non-specific band in #1, darkening these bands relative to the others. See the original, uncropped blot to best appreciate this. A comment is now in the figure legend.

Reviewer #2: This very interesting article by Cho et al., describes Pk isoforms Pksple and Pkpk's expression and function during planar polarization of the Drosophila eye. The authors provide a detailed analysis of the expression patterns on Pk isoforms, including the undescribed Pkm isoform. The authors propose that the amount of Pkpk seen in Pk sple mutant would be more than enough to prevent mistakes in R3/R4 cell fate decisions although it can´t orient them along the dorsal-ventral axis. On the other hand both Pkpk and Pksple can sustain proper ommatidia rotation. Finally the authors propose that Pkpk competes with Pksple for localization at the apical membrane.

Major comment

Although the article is well structured and provides clarifying experiments regarding Pk function in the eye, it lacks replicates and statistical analyses. The authors do not mention whether the observations are made based on one case or different biological replicates. They do not provide statistical analysis or quantification of stainings and phenotypes. It could be understood to confirm well-described observations, but one expects new observations based on more quantitative data, for instance, quantification of immunostaining of Pk in the pksple background or Pksple in a pkpk background. Also, ommatidia maturation based on Vang staining.

I believe this is an essential part of the study, and it should be better characterized.

Similarly, phenotypic observations in the vang rescue experiment seem to be based only on one eye section, clonal analysis in one clone, and western blots do not appear to have replicates.

Response: Biological replicates: numbers of biological replicates are now indicated in the manuscript.

Phenotypic quantifications are now provided in the manuscript (lines 248-251, 276-277, 346, Figure S7 legend). 

Quantification of immunostaining: We agree that proper quantification can sometimes provide better insight about relationships between molecular changes and their phenotypic effects. Here, we aim to show varying levels of a given isoform in different ages of ommatidia within the same specimen. In this case, the precise magnitude of the changes is not important to our arguments. Furthermore, comparisons within a specimen and especially between different specimens suffer from technical limitations including that detection may be non-linear, antibodies may saturate, and background staining may vary. We have therefore not attempted to quantify these changes. 

We’re not sure what the referee is asking about with respect to ommatidial maturation based on Vang staining. Vang patterns during ommatidial maturation have been analyzed previously by Tanya Wolff (Rawls and Wolff, 2003 Development), and we have used the previously described development of asymmetric localization as a marker of polarization in this manuscript. Arrows have been added to Figure 3 to better highlight this asymmetry.

Quantification of western blots: CRISPR knock-out for pkpk-sple and pkm, or knock-in for pkm in S6-2 B, C figs gave all or none band patterns confirming intended chromosome engineering and quantification is not necessary. To analyze the dependency of one isoform on the other in terms of the amount of protein, we monitored Pkpk or Pksple in pksple or pkpk mutant backgrounds, respectively. For wings, we prepared protein lysates from four different developmental stages (8, 20, 36, 40H apf) although here we are only showing the results from 36 and 40H apf when both Pkpk and Pksple isoforms are similarly expressed. From all stages, we did not observe meaningful differences in the levels of isoforms depending on presence or absence of the other. Had we observed differences of a potentially meaningful magnitude, we would have made efforts to better quantify them. We now provide relative band intensities for the levels we saw. 

Minor comments

Calibrations bars are missing in all figures

Response: Scale bars were added

legend figure 3: the authors claim that there is colocalization , however there is no colocalization analysis. Also they claim that the asymmetry increases, which is not clear from the picture. The authors should provide a quantitative analysis that supports their observations and indicate the asymmetry using arrows or arrowheads.

Response: It was shown that Vang physically interacts with Pk and our intention is to show that Pk[sple] is the isoform participating in the early ommatidial R3/4 pairs. Colocalization has previously been shown, and we respectfully submit that this is evident from the overlap of red and green signal producing yellow in the image. We altered the figure legend from “asymmetry increase” to “showing asymmetry” so as not to argue here that there is a quantitative increase in asymmetry, though we think this is abundantly clear from previously published observations. Arrows were added to indicate the asymmetric Vang and Pksple in R3/R4 pairs.

Line 185-186: Authors should indicate figure 2 when they mentioned that endogenously tagged Pkpk in wt ommatidia becomes highly expressed around row 9.

Response: Done

line 262: missing the number of the suppl figure (Fig 5 and S Fig)

Response: Corrected

6. PLOS authors have the option to publish the peer review history of their article (what does this mean?). If published, this will include your full peer review and any attached files.

Do you want your identity to be public for this peer review? For information about this choice, including consent withdrawal, please see our Privacy Policy.

Reviewer #1: No

Reviewer #2: No

---

## [Decision Letter · Decision Letter 1]

22 Dec 2021

Prickle isoform participation in distinct polarization events in the Drosophila eye

PONE-D-21-29540R1

Dear Dr. Axelrod,

We’re pleased to inform you that your manuscript has been judged scientifically suitable for publication and will be formally accepted for publication once it meets all outstanding technical requirements.

Kind regards,

Carlos Oliva, PhD

Academic Editor

PLOS ONE

Additional Editor Comments (optional):

Reviewers' comments:

Reviewer's Responses to Questions

**Comments to the Author**

1. If the authors have adequately addressed your comments raised in a previous round of review and you feel that this manuscript is now acceptable for publication, you may indicate that here to bypass the “Comments to the Author” section, enter your conflict of interest statement in the “Confidential to Editor” section, and submit your "Accept" recommendation.

Reviewer #2: All comments have been addressed

2. Is the manuscript technically sound, and do the data support the conclusions?

Reviewer #2: Yes

3. Has the statistical analysis been performed appropriately and rigorously? 

Reviewer #2: N/A

4. Have the authors made all data underlying the findings in their manuscript fully available?

Reviewer #2: Yes

5. Is the manuscript presented in an intelligible fashion and written in standard English?

Reviewer #2: Yes

6. Review Comments to the Author

Reviewer #2: (No Response)

7. PLOS authors have the option to publish the peer review history of their article (what does this mean?). If published, this will include your full peer review and any attached files.

Reviewer #2: No

---

## [Editor Report · Acceptance letter]

3 Feb 2022

PONE-D-21-29540R1 

Prickle isoform participation in distinct polarization events in the *Drosophila* eye 

Dear Dr. Axelrod:

I'm pleased to inform you that your manuscript has been deemed suitable for publication in PLOS ONE. Congratulations! Your manuscript is now with our production department. 

Kind regards, 

on behalf of

Dr. Carlos Oliva 

Academic Editor

PLOS ONE